# Bayesian Joint Estimation of Multiple Graphical Models

**Lingrui Gan,  Xinming Yang,  Naveen N. Nariestty,  Feng Liang**
Department of Statistics
University of Illinois at Urbana-Champaign
{lgan6, xyang104, naveen, liangf}@illinois.edu

## Abstract

In this paper, we propose a novel Bayesian group regularization method based on the spike and slab Lasso priors for jointly estimating multiple graphical models. The proposed method can be used to estimate common sparsity structure underlying the graphical models while capturing potential heterogeneity of the precision matrices corresponding to those models. Our theoretical results show that the proposed method enjoys the optimal rate of convergence in $\ell_\infty$ norm for estimation consistency and has a strong structure recovery guarantee even when the signal strengths over different graphs are heterogeneous. Through simulation studies and an application to the capital bike-sharing network data, we demonstrate the competitive performance of our method compared to existing alternatives.

## 1 Introduction

Gaussian graphical models (GGMs) are widely studied from both the frequentist [30, 9, 3, 18, 28] and the Bayesian perspectives [4, 7, 26, 1, 19, 10, 14]. A GGM model assumes that a collection of variables jointly follows a multivariate Gaussian distribution with an unknown precision matrix. It is well known that there is a one-to-one correspondence between the sparsity pattern of the precision matrix of a Gaussian distribution and the graph that describes the conditional dependence structure among the variables: Non-zero entries in the precision matrix correspond to edges in the graph [6]. Due to this connection, given data from a GGM, we are interested in estimating not only the precision matrix but also its support.

In many applications, observations are naturally grouped into different classes. For example, in biological experiments, subjects are classified into categories based on their experimental conditions; in social network data, users are grouped by users' characteristics; and in gene expression analysis, expression data are classified into different tissues or disease states. In such situations, it is restrictive to assume that all observations follow the same graphical model, i.e., have the same precision matrix, and it would be more suitable to assume that different classes have different precision matrices. Meaningful insights can be more effectively extracted, if we utilize the cross-class similarities of the precision matrices and estimate graphs for the multiple classes jointly.

Several approaches have been proposed for jointly estimating multiple GGMs. From the penalized likelihood perspective, [12, 5, 13, 17] extended approaches for estimating a single graph to the multiple-graph setting by introducing group-level penalty terms and studied the theoretical properties of these approaches. From the Bayesian perspective, Peterson et al. [21] proposed a Markov random field prior on multiple graphs to encourage the selection of common edges in related graphs. Tan et al. [25] proposed to use a Chung-Lu random graph model as the prior for hierarchical modeling of multiple GGMs. However, theoretical guarantees of the Bayesian methods are not available.

In this paper, we develop a Bayesian approach for jointly estimating multiple GGMs under the assumption that the multiple precision matrices share a common sparsity structure but they can have heterogeneous signal magnitudes. We provide theoretical results showing that the maximum a posteriori (MAP) estimators have the optimal rate of convergence in $\ell_\infty$ norm, even under model mis-specification when precision matrices for different classes do not share a common sparsity structure. When the multiple GGMs do share a common sparsity structure, the proposed approach is proved to be consistent in recovering such a structure, even when some of the within group signals are weaker than the $(\sqrt{(\log p)/n})$ rate, the minimal signal strength usually required for consistency results on structure recovery .

The remaining part of the paper is organized as follows. The Bayesian formulation and parameter estimation procedure for our model are provided in Section 2. Theoretical guarantees of our approach are presented in Section 3 and empirical studies are provided in Section 4.

## 2 Method

Let $Y_1, \ldots, Y_K$ denote the data from $K$ classes, where the $k$-th dataset $Y_k$ consists of $n_k$ observations $(Y_{k,1}, \ldots, Y_{k,n_k})$, each of which is a $p$-dimensional vector. Throughout we assume the $p$ variables are common across the $K$ classes and data from each class follow a $p$-dimensional Gaussian distribution:

$$Y_{k,1}, \ldots, Y_{k,n_k} \sim N_p(\mathbf{0}, \Theta_k^{-1}).$$

Our target is to estimate the $K$ precision matrices $\Theta_k = (\theta_{k,ij})$, $k \in \{1, \ldots, K\}$, and recover their common sparsity structure.

### 2.1 Bayesian Formulation

For regularized estimation and sparsity recovery, we shall place prior distributions on the $\Theta_k$'s by first introducing binary latent variables $\gamma_{ij}$ which indicate whether nodes $i$ and $j$ have an edge ($\gamma_{ij} = 1$) or not ($\gamma_{ij} = 0$). A Bernoulli prior on $\gamma_{ij}$ and a spike and slab prior on the off-diagonal elements of $\Theta_k$ are placed as follows:

$$p(\theta_{k,ij}|\gamma_{ij}) = \begin{cases} \text{LP}(\theta_{k,ij}; v_1) \text{ when } \gamma_{ij} = 1, \\ \text{LP}(\theta_{k,ij}; v_0) \text{ when } \gamma_{ij} = 0, \end{cases} \quad \gamma_{ij} \sim \text{Bern}(p_1), \tag{2.1}$$

where $v_1 > v_0 > 0$ and $\text{LP}(\cdot; v)$ is the Laplace distribution with parameter $v$. For $\gamma_{ij} = 1$, $\theta_{k,ij}$'s represent signals modeled by a slab distribution with high variance, and for $\gamma_{ij} = 0$, $\theta_{k,ij}$'s represent noise modeled by a spike distribution with mass tightly centered around zero. Integrating over $\gamma_{ij}$, we have the following multivariate spike and slab prior on the group of $K$ entries $\boldsymbol{\theta}_{ij} = (\theta_{1,ij}, \ldots, \theta_{K,ij})$:

$$p(\boldsymbol{\theta}_{ij}) = p_1 \prod_{k=1}^{K} \text{LP}(\theta_{k,ij}; v_1) + (1 - p_1) \prod_{k=1}^{K} \text{LP}(\theta_{k,ij}; v_0). \tag{2.2}$$

When $K = 1$, the distribution above is the one-dimensional spike and slab Lasso prior utilized for linear regression by [23, 24] and for analyzing a single GGM by [10].

We impose the aforementioned spike and slab Lasso prior (2.2) on the upper triangular of the precision matrices, i.e., on $\boldsymbol{\theta}_{ij}$ for $i < j$, and enforce $\boldsymbol{\theta}_{ji} = \boldsymbol{\theta}_{ij}$ to keep $\Theta_k$ symmetric. In addition, we place independent exponential priors on the positive diagonal entries to introduce a small shrinkage. In summary, the Bayesian prior formulation of our model is as follows:

$$\begin{aligned} \theta_{k,ii} &\sim \text{Exp}(\tau), \\ \boldsymbol{\theta}_{ij} &\sim p_1 \prod_{k=1}^{K} \text{LP}(\theta_{k,ij}; v_1) + (1 - p_1) \prod_{k=1}^{K} \text{LP}(\theta_{k,ij}; v_0), \end{aligned} \tag{2.3}$$

where $i < j$ and $k = 1, \ldots, K$.

### 2.2 Parameter Estimation

We first focus on the point estimation of $\boldsymbol{\Theta} = (\Theta_1, \ldots, \Theta_K)$ and then discuss the posterior inference of the sparsity structure conditional on the point estimator (discussed in Section 2.3). Motivated

by [16, 10], we estimate $\boldsymbol{\Theta}$ by solving the following optimization problem under the constraint of $\Omega = \{\boldsymbol{\Theta} : \Theta_k > 0, \|\Theta_k\|_2 \leqslant B, \ k = 1, \ldots, K\}$ where $\Theta_k > 0$ indicates that $\Theta_k$ is positive definite and $\|\Theta_k\|_2$ denotes the spectral norm of $\Theta_k$. Our constrained MAP estimator is given by

$$\hat{\boldsymbol{\Theta}} = \underset{\boldsymbol{\Theta} \in \Omega}{\arg\min} \left( -\log p(\boldsymbol{Y} \mid \boldsymbol{\Theta}) + \alpha \mathrm{Pen}(\boldsymbol{\Theta}) \right), \quad \alpha \geqslant 1, \tag{2.4}$$

where $\log p(\boldsymbol{Y} \mid \boldsymbol{\Theta})$ is the log likelihood and $\mathrm{Pen}(\boldsymbol{\Theta})$ is the negative log of the prior on $\boldsymbol{\Theta}$:

$$\begin{cases} \log p(\boldsymbol{Y} \mid \boldsymbol{\Theta}) = \sum_{k=1}^{K} \frac{n_k}{2} \left( \log \det(\Theta_k) - \mathrm{tr}(S_k \Theta_k) \right), \quad S_k = \frac{1}{n_k} \sum_{i=1}^{n_k} Y_{k,i} Y_{k,i}^T, \\ \mathrm{Pen}(\boldsymbol{\Theta}) = \sum_{i=1}^{p} \sum_{k=1}^{K} \tau \theta_{k,ii} + \sum_{i<j} -\log \left( \frac{p_1}{(2v_1)^K} e^{-\frac{\|\boldsymbol{\theta}_{ij}\|_1}{v_1}} + \frac{1 - p_1}{(2v_0)^K} e^{-\frac{\|\boldsymbol{\theta}_{ij}\|_1}{v_0}} \right), \end{cases}$$

with $S_k$ being the sample covariance matrix of the $k$-th class and $\|\boldsymbol{\theta}_{ij}\|_1$ being the $\ell_1$ norm of the vector $\boldsymbol{\theta}_{ij}$. From the Bayesian viewpoint, our estimator (2.4) is equivalent to the MAP estimator of the posterior $p(\boldsymbol{\Theta} \mid \boldsymbol{Y}) \propto p(\boldsymbol{Y} \mid \boldsymbol{\Theta}) p(\boldsymbol{\Theta})^\alpha$, where the prior is raised to the power of $\alpha$ so that its influence on inference can be appropriately magnified. From the penalized likelihood viewpoint, we are essentially multiplying the penalty function $\mathrm{Pen}(\boldsymbol{\Theta})$ by a multiplier of $\alpha$, which is equivalent to scaling the log likelihood by $1/\alpha$. Such an adjustment is commonly adopted to develop optimal theoretical results, for example, by [8, 31, 16].

For scalability, we propose to compute the MAP estimator instead of sampling from the full posterior. Full posterior sampling for high-dimensional GGMs is computationally expensive, for example, in [21, 25], the dimension $p$ in all empirical studies is at most 22 due to the computational limitations. Although we only propose a point estimator (2.4) for the precision matrices, our model is still formulated from a Bayesian perspective with a continuous spike and slab prior distribution. While this prior does not directly place mass on sparse solutions as in [27, 29], the latent binary indicators $\gamma_{ij}$ introduced can distinguish between "signal" and "noise". This is a common technique used in the Bayesian literature [11, 20, 23, 10] to avoid the computational bottleneck of degenerate priors. Furthermore, with the Bayesian machinery, we are able to extract the posterior inclusion probabilities for structure recovery in Section 2.3 and provide strong guarantees for graph selection in Section 3.3.

The term $\mathrm{Pen}(\boldsymbol{\Theta})$, which is induced from our prior specification, acts as a non-convex penalty function. The non-convexity of the penalty brings desired shrinkage effects, as shown in our theoretical results in Section 3 and prior results in the literature [31, 16, 15]. However, it may cause the whole objective function to be non-convex, and consequently, we need to deal with multiple local solutions to the minimization problem (2.4). In the following theorem, we show that when being constrained to the parameter space $\Omega$, the minimization problem (2.4) is in fact strictly convex. Thus, the solution $\hat{\boldsymbol{\Theta}}$ of the objective function (2.4) will be unique. Proof of this result is in Supplementary Material.

**Theorem 1** *If* $B < \sqrt{\frac{2nv_0^2}{\alpha K}}$, *then the constrained minimization problem* (2.4) *is strictly convex.*

**Remark.** The upper bound on $B$ can increase with sample size $n$. When we establish the selection consistency of $\hat{\boldsymbol{\Theta}}$ in Theorem 2, we will require the order of $v_0^2$ to be $O\left(\alpha^2/(n\log p)\right)$. Therefore, the upper bound on $B$ is $O\left(\sqrt{\alpha/(K\log p)}\right)$, which can go to $+\infty$ as long as the order of $\alpha$ is greater than $K\log p$.

## 2.3 Common Structure Recovery

Utilizing the hierarchical structure (2.1) of the spike and slab prior, we make inference on the common sparsity structure based on the following posterior inclusion probability (PIP)

$$\mathbb{P}(\gamma_{ij} = 1 \mid \hat{\boldsymbol{\theta}}_{ij}) = \frac{1}{1 + \frac{1-p_1}{p_1} \left(\frac{v_1}{v_0}\right)^K \exp\left\{ -\left(\frac{1}{v_0} - \frac{1}{v_1}\right) \|\hat{\boldsymbol{\theta}}_{ij}\|_1 \right\}} \triangleq p(\hat{\boldsymbol{\theta}}_{ij}). \tag{2.5}$$

We can estimate the common sparsity structure by thresholding the PIP, e.g., with $t = 1/2$:

$$\hat{\mathcal{S}} = \left\{ (i,j) : p(\hat{\boldsymbol{\theta}}_{ij}) > t, \ \text{for } t \in (0,1) \right\}. \tag{2.6}$$

Note that the PIP is a function of $\|\hat{\boldsymbol{\theta}}_{ij}\|_1$, so that the signal strength in the whole group is utilized together to estimate the common sparsity pattern. Even when some individual entries are of small signal strength, the information shared within its group could help us to identify the shared structure. Our theoretical results provided in Section 3 confirm that this strategy will be indeed beneficial for recovering such signals.

## 3 Theoretical Guarantees

In this section, we develop the theoretical properties of the proposed estimator $\hat{\Theta}$ including estimation accuracy and structure recovery consistency. For simplicity, we assume the sample sizes of the $K$ classes are the same with $n_1 = \cdots = n_K = n$ in the theoretical analysis.

**Notations:** For a square matrix $A_{p \times p} = (a_{ij})$, we denote its element-wise $\ell_\infty$ norm by $\|A\|_\infty = \max_{1 \leqslant i,j \leqslant p} |a_{ij}|$; its Frobenius norm by $\|A\|_F$; and its spectral norm by $\|A\|_2$. We denote the largest eigenvalue and smallest eigenvalue of $A$ by $\lambda_{\max}(A)$ and $\lambda_{\min}(A)$, respectively. When $A$ is a square symmetric matrix, we note $\|A\|_2 = \lambda_{\max}(A)$. For a collection of $K$ square matrices of the same dimension $\boldsymbol{A} = (A_1, \ldots, A_K)$, write $\|\boldsymbol{A}\|_\infty = \sup_{1 \leqslant k \leqslant K} \|A_k\|_\infty$. Let $\boldsymbol{\Theta}^0 = (\Theta_1^0, \ldots, \Theta_K^0)$ denote the collection of true precision matrices and $\mathcal{S}_k^0 = \{(i,j) : \theta_{k,ij}^0 \neq 0\}$ denote the index set of nonzero entries in the true precision matrix $\Theta_k^0$. Define column sparsity of $\Theta_k^0$ as $d_k = \max_i card(\{j : \theta_{k,ij}^0 \neq 0\})$ where $card(\cdot)$ denotes the cardinality of a set and let $d = \max_k d_k$.

### 3.1 Conditions

In our theoretical analysis, we do not restrict the observed data to follow a Gaussian distribution. Thus, our Bayesian hierarchical model (2.3) could be treated as a working model. The observed data are allowed to be from any distribution with exponential tails (e.g., sub-Gaussian distributions) or polynomial tails (e.g., $t$ distributions), which is the same setup considered in [3, 10] when the class size $K = 1$. Specifically, for all the $p$-dimensional random vectors $Y_{k,i} = (Y_{k,i}^{(1)}, \ldots, Y_{k,i}^{(p)})$, $i = 1, \ldots, n_k$ and $k = 1, \ldots, K$, we have the following assumptions:

(A.1) Exponential tail condition: there exist some constants $0 < \eta < 1/4$ and $U > 0$ such that $(\log p)/n < \eta$ and

$$\mathbb{E}(e^{tY_{k,i}^{(j)}}) \leqslant U \quad \text{for any } |t| \leqslant \eta \text{ and } j = 1, \ldots, p; \qquad (3.1)$$

(A.2) Polynomial tail condition: there exist some constants $\kappa_1, \kappa_2, \kappa_3, U > 0$ such that $p \leqslant \kappa_1 n^{\kappa_2}$ and

$$\mathbb{E}|Y_{k,i}^{(j)}|^{4+4\kappa_2+\kappa_3} \leqslant U \quad \text{for } j = 1, \ldots, p. \qquad (3.2)$$

We shall establish estimation and selection consistency of our method when the true data distribution satisfies (A.1) or (A.2) and is not necessarily a multivariate Gaussian distribution. Note that when the data indeed follows a multivariate Gaussian distribution, (A.1) is satisfied.

In addition, we assume that the eigenvalues of the true precision matrices are bounded:

(A.3) Eigenvalue condition: $1/\xi_0 \leqslant \lambda_{\min}(\Theta_k^0) \leqslant \lambda_{\max}(\Theta_k^0) \leqslant 1/\xi_1$ for $k = 1, \ldots, K$.

### 3.2 Estimation Accuracy

The following theorem establishes the rate of convergence of the proposed estimator under $\ell_\infty$ norm. For this result, we do not require the different precision matrices to have the same sparsity structure.

**Theorem 2** *Suppose one of the tail conditions, (A.1) or (A.2), holds and the true precision matrices satisfy (A.3). Let $C_1 = \eta^{-1}(2 + \kappa_0 + \eta^{-1}U^2)$ when the exponential tail condition (A.1) holds and $C_1 = \sqrt{(\|\boldsymbol{\Theta}^0\|_\infty + 1)(4 + \kappa_0)}$ when the polynomial tail condition (A.2) holds for some $\kappa_0 > 0$. In addition, assume that*

*(i) the hyperparameters $(v_1, v_0, p_1, \tau)$ satisfy:*

$$\begin{cases} \max(\frac{3}{v_1}, 2\tau) < C_3 \sqrt{\frac{n \log p}{\alpha^2}}, \frac{1}{v_0} > C_4 \sqrt{\frac{n \log p}{\alpha^2}}, \\ \epsilon_2 < \frac{v_1^K(1-p_1)}{v_0^K p_1} \leqslant \frac{v_1^{K+2}(1-p_1)}{v_0^{K+2} p_1} \leqslant 2p^{\epsilon_0/\alpha}; \end{cases}$$

*(ii) the sample size $n$ satisfies:* $\sqrt{n} \geqslant M_0 \max(d, \sqrt{K})\sqrt{\log p}$;

*(iii) the bounds on the spectral norms of the estimated precision matrices satisfy:*

$$1/\xi_1 + dC_5\sqrt{\frac{\log p}{n}} < B < \left(\frac{2nv_0^2}{\alpha K}\right)^{1/2};$$

*(iv) the parameter $\alpha$ satisfies:* $\alpha p^{\epsilon_0/\alpha} > KC_3^2 \log p/(2\xi_1^2)$.

*Then, the minimizer $\hat{\Theta}$ is unique and satisfies*

$$\|\hat{\Theta} - \Theta^0\|_\infty < C_5\sqrt{\frac{\log p}{n}},$$

*with probability greater than $1 - K\delta$, where $\delta = 2p^{-\kappa_0}$ when condition (A.1) holds, and $\delta = O(n^{-\kappa_3/8} + p^{-\kappa_0/2})$ when condition (A.2) holds. Moreover, $\left(\hat{\Theta}_k\right)_{ij} = 0$ for $(i,j) \in \left(\mathcal{S}_k^0\right)^c$. Here, $C_3, \epsilon_2$ are sufficiently small positive constants, $M_0, C_4, C_5, \epsilon_0$ are positive constants only depend on the ground truth $\Theta^0$.*

Our proof is motivated by the constructive proof technique used in [22] and [10]. Details of the definitions of $M_0, C_4, C_5, \epsilon_0$ and the proof of Theorem 2 are provided in Supplementary Material.

Condition (i), which is related to the rates of the hyperparameters, controls the level of shrinkage of the penalty function $\text{Pen}(\Theta)$. With a proper choice of the hyperparameters, our penalty function induces an appropriate adaptive shrinkage effect: the shrinkage is strong enough when the magnitude of $\theta$ is small to kill the noise and produce exact zero, and is insignificant when the magnitude of $\theta$ is large so that the bias is controlled. Condition (ii) is on the relationship between the sample size $n$ and the number of variables $p$, and $p$ could grow nearly exponentially with $n$. Condition (iii) deals with the parameter space of the constrained optimization problem, which ensures both the feasibility and convexity of the problem. Under these conditions, our Theorem 2 states that, as long as the parameter $\alpha$ satisfies the condition (iv), the error rate of every entry of the estimated precision matrices is at most $O_p(\sqrt{(\log p)/n})$.

### 3.3 Sparsity Structure Recovery Consistency

Besides the estimation accuracy, another important task is to identify the sparsity structure of the precision matrices as it tells the conditional dependence relationships between the $p$ variables of interest. If the minimal signal strength satisfies $\min_k \min_{i \neq j, (i,j) \in \mathcal{S}_k^0}(|\theta_{k,ij}^0|) > L_0\sqrt{(\log p)/n}$ for some sufficiently large constant $L_0$, Theorem 2 directly gives rise to the result that our estimator $\hat{\Theta}_k$ has the same sparsity structure as the truth $\mathcal{S}_k^0$ with probability converging to 1, even when different classes do not have the same sparsity structure. If all precision matrices share a common sparsity structure, i.e., $\mathcal{S}_1^0 = \cdots = \mathcal{S}_K^0 = \mathcal{S}^0$, then our proposed method achieves selection consistency with a weaker condition on the minimal signal strength as stated in the following theorem.

**Theorem 3** *Suppose conditions in Theorem 2 all hold. In addition, assume that:*

*(v) the minimal signal strength satisfies*

$$\min_{(i,j) \in \mathcal{S}^0}(\max_k |\theta_{k,ij}^0|) \geqslant L_0\sqrt{(\log p)/n},$$

*where $L_0 > C_5$ is a sufficiently large constant;*

*(vi) rates of the hyperparameters $v_1, v_0$, and $p_1$ satisfy*

$$\frac{1-p_1}{p_1}\left(\frac{v_1}{v_0}\right)^K \geqslant \frac{1-t}{t} > \frac{\frac{1-p_1}{p_1}\left(\frac{v_1}{v_0}\right)^K}{p^{(C_4-C_3)(L_0-C_5)/\alpha}},$$

*where $t$ is an arbitrary thresholding value between $0$ and $1$. Then we have*

$$\mathbb{P}(\hat{\mathcal{S}} = \mathcal{S}^0) \to 1.$$

Condition (v) is the condition on the minimal signal strength. Compared to similar conditions required by approaches that estimate each GGM individually, this condition is weaker since it only places requirements on the largest signal within each group. Therefore, the whole group would benefit from one large signal. Under the weaker minimal signal strength condition and with appropriate choice of the hyperparameters satisfying condition (vi), we can differentiate between the "signal" and "noise" groups with probability going to 1. A proof of Theorem 3 is provided in the Supplementary Material.

## 3.4 Comparisons with Existing Works

In this section, we compare our theoretical results in estimation accuracy and selection consistency with other alternatives [12, 13]. In the following discussion, we use $\tilde{\Theta}$ as a generic notation to denote estimators proposed by others.

Guo et al. [12] established the estimation accuracy of their estimator $\tilde{\Theta}$ in Frobenius norm for a fixed $K$ value: $\sum_{k=1}^{K} \|\tilde{\Theta}_k - \Theta_k^0\|_F = O_p\left(\sqrt{\frac{(p+q_1)\log p}{n}}\right)$, where $q_1 = card(\cup_k\{\mathcal{S}_k^0\}) - p$. We note that our Theorem 2 gives rise to the same rate as theirs under Frobenius norm. For recovering the graph structure, Guo et al. [12] obtained sparsistency, i.e., the zero entries in the true precision matrices are estimated as zeroes with probability tending to one. However, there is no guarantee that the nonzero entries could be detected. This is weaker than our Theorem 3 as we recover the entire graph structure. Moreover, to achieve sparsistency, Guo et al. [12] require the minimum signal strength $\min_k \min_{i\neq j,(i,j)\in\mathcal{S}_k^0}(|\theta_{k,ij}^0|)$ to be lower bounded by some constant while we allow it to go to zero.

Lee and Liu [13] established the estimation accuracy of their estimator $\tilde{\Theta}$ in the averaged version of the $\ell_\infty$-$\ell_1$ norm: $\max_{i,j}\left(\frac{1}{K}\sum_{k=1}^{K}\left|\tilde{\theta}_{k,ij} - \theta_{k,ij}^0\right|\right) = O_p\left(\sqrt{\frac{\log p}{n}}\right)$. Our estimation error rate from Theorem 2 is on the maximum over all entries of all precision matrices without averaging, and therefore is stronger. In particular, their result is a direct consequence of ours. For selection consistency, the major difference between theirs and ours is the condition on the signal strength. Lee and Liu [13] implicitly require $\min_k \min_{i\neq j,(i,j)\in\mathcal{S}_k^0}(|\theta_{k,ij}^0|)$ to be lower bounded at the rate of $K(\log p/n)^{1/2}$, where $K$ is the class size, while we only require a smaller signal strength $(\log p/n)^{1/2}$. In addition, our requirement is on the lower bound of $\min_k \max_{i\neq j,(i,j)\in\mathcal{S}_k^0}(|\theta_{k,ij}^0|)$, which is weaker than requirement on the lower bound of $\min_k \min_{i\neq j,(i,j)\in\mathcal{S}_k^0}(|\theta_{k,ij}^0|)$.

# 4 Numerical Studies

## 4.1 Computation: an EM Algorithm

For computation, we propose an EM algorithm by treating $\Gamma = (\gamma_{ij})$ as latent variables and estimating $\Theta$ by applying the following two steps iteratively:

- E-step: Calculate the posterior distribution $\mathbb{P}(\gamma_{ij} = 1 \mid \Theta^{(t)}) := p_{ij}(\theta_{ij}^{(t)})$, which follows the formula in (2.5), and compute the so-called $Q$ function, the expectation of the full log-likelihood with respect to $\mathbb{P}(\gamma_{ij} = 1 \mid \Theta^{(t)})$:

$$Q(\Theta) = \sum_{k=1}^{K}\left\{\frac{n_k}{2\alpha}\left(\log\det(\Theta_k) - \text{tr}(S_k\Theta_k)\right) - \sum_{i=1}^{p}\tau\theta_{k,ii} - \sum_{i<j}\left[\frac{p_{ij}(\theta_{ij}^{(t)})}{v_1} + \frac{1 - p_{ij}(\theta_{ij}^{(t)})}{v_0}\right]|\theta_{k,ij}|\right\}.$$

- M-step: The $Q$ function is a summation of $K$ terms with each to be a weighted graphical Lasso [9] problem. Therefore, in the M-step, we maximize the $Q$ function within in the parameter space $\Omega$, utilizing algorithms for graphical Lasso. As a result, the computational complexity of our EM algorithm is $O(p^3)$, which is as efficient as the state-of-the-art algorithms for graphical Lasso problems [9, 10].

Derivations and implementation details of the algorithm are provided in the Supplementary Material.

## 4.2 Simulation Results

Following the simulation setups in [12, 5, 21, 13], we assess the performance of our proposed method under six different settings: three nearest-neighbor networks and three scale-free networks. The details of the settings are described as follows.

1. Nearest-neighbor network: we randomly generate $p$ points on a unit square and find the $o$ nearest neighbors of each point in terms of the Euclidean distance. The baseline nearest-neighbor network is constructed by linking any two points which are the $o$-nearest neighbors of each other. Larger $o$ induces a denser network and here, we use $o = 3$. After that, we generate $K$ individual networks by adding $\rho M$ individual edges to the baseline graph with $M$ to be number of edges in the baseline graph and $\rho = 0, 0.25, 0.5$.

   Given a network structure, we generate the corresponding precision matrix $\Theta_k$ by assigning ones on diagonal entries, zeros on entries not corresponding to network edges, and values from a uniform distribution with support on $[-1, -0.5] \cup [0.5, 1]$ on entries corresponding to edges. To ensure positive definiteness, we then divide each off-diagonal element $\theta_{k,ij}$ by $1.01\sqrt{\sum_{i:i \neq j}|\theta_{k,ij}|}\sqrt{\sum_{j:j \neq i}|\theta_{k,ij}|}$.

2. Scale-free network: many real-world large networks, such as the world wide web, social networks, and collaboration networks, are thought to be scale-free. We construct the baseline scale-free network using the Barabási-Albert model [2]. Next, individual networks and corresponding precision matrices are generated in the same way as in the first design.

In each setting, we set $K = 3$ and $p = n_k = 100$, and, for each $k \in \{1, \ldots, K\}$, we generate $n_k$ independently and identically distributed observations from a multivariate Gaussian distribution with mean $\mathbf{0}$ and precision matrix $\Theta_k$. We compare our method with $\alpha = 1$ and $\alpha = n$ with three different methods: fitting each class individually by BAGUS (denoted as BAGUS) [10]; ignoring the class information and fitting a single model by BAGUS (denoted as Pooled); the group graphical Lasso (denoted as GGL) [5]. Bayesian approaches based on full posterior sampling [21, 25] are not considered for comparison as their Markov chain Monte Carlo (MCMC) samplers are not scalable with large $p$. For all methods, we use a grid search to select the set of hyperparamters that minimizes BIC. For BAGUS and Pooled methods, we follow the same tuning procedure in [10] and tune the spike and slab prior parameters $(v_0, v_1)$ with $v_0 = (0.25, 0.5, 0.75, 1) \times \sqrt{1/(n \log p)}$ and $v_1 = (2.5, 5, 7.5, 10) \times \sqrt{1/(n \log p)}$. For GGL, we tune the two penalty parameters $(\lambda_1, \lambda_2)$ as in [5] with $\lambda_1 = (0.1, 0.2, \ldots, 1)$ and $\lambda_2 = (0.1, 0.3, 0.5)$.

To compare the performance of the methods, we calculate specificity (Spec), sensitivity (Senc), Matthews correlation coefficient (MCC), area under the ROC curve (AUC), Frobenius norm (F-norm), and element-wise $\ell_\infty$ norm ($\ell_\infty$ norm) for each class. In Table 1-2, we report the maximum of $\ell_\infty$ norm and the average of the other measures over the $K$ classes and the results are aggregated based on 100 replications. From the results, we observe that our method performs the best in all the designs in terms of both selection accuracy (MCC and AUC) and estimation accuracy (F-norm and $\ell_\infty$ norm). Even when $\rho \neq 0$, that is, the sparsity patterns over classes are different, which deviates from our assumption, our method still has the best performance.

The average computational times of all the methods using a MacBook Pro with 2.9 GHz Intel Core i5 processor and 8.00 GB memory are reported in Table 3. The computational time of our method is comparable to the competitors except the Pooled method, which restrictively assumes the same precision matrix for all classes and has much worse performance compared to our method. Therefore, our method is competitive even after considering the runtimes.

## 4.3 Application to Capital Bikeshare Data

We use Capital Bikeshare trip data[1] to evaluate the performance of the proposed method. The data contains records of bike rentals in a bicycle sharing system with more than $500$ stations. We consider $p = 237$ stations located in Washington, D.C. and record the number of rentals started at these stations for every day in 2016, 2017 and 2018. Following the same processing procedure in [32], we remove the seasonal trend and marginally transform each station's data to a normal distribution.

Table 1: Result of nearest-neighbor network

| | Spec | Sens | MCC | AUC | F-norm | $\ell_\infty$ norm |
|---|---|---|---|---|---|---|
| | | | $n = 100$, $p = 100$, $\rho = 0$ | | | |
| Our method ($\alpha = 1$) | 1.000(0.000) | 0.920(0.038) | 0.955(0.022) | 0.974(0.017) | 2.576(0.221) | 0.503(0.083) |
| Our method ($\alpha = n$) | 1.000(0.000) | 0.993(0.008) | **0.991(0.009)** | **0.996(0.004)** | **2.094(0.150)** | **0.449(0.099)** |
| BAGUS | 0.994(0.002) | 0.816(0.039) | 0.794(0.033) | 0.903(0.022) | 3.184(0.190) | 0.551(0.093) |
| Pooled | 0.989(0.003) | 0.664(0.056) | 0.616(0.048) | 0.840(0.029) | 7.115(0.380) | 0.983(0.035) |
| GGL | 0.948(0.008) | 0.707(0.074) | 0.401(0.044) | 0.845(0.038) | 6.338(0.382) | 0.604(0.037) |
| | | | $n = 100$, $p = 100$, $\rho = 0.25$ | | | |
| Our method ($\alpha = 1$) | 0.994(0.001) | 0.823(0.034) | 0.803(0.015) | **0.954(0.014)** | **2.862(0.145)** | **0.443(0.066)** |
| Our method ($\alpha = n$) | 0.992(0.003) | 0.889(0.021) | **0.823(0.025)** | 0.943(0.010) | 2.867(0.154) | 0.443(0.074) |
| BAGUS | 0.988(0.003) | 0.813(0.030) | 0.732(0.025) | 0.917(0.017) | 3.372(0.148) | 0.591(0.102) |
| Pooled | 0.976(0.004) | 0.571(0.045) | 0.472(0.029) | 0.783(0.024) | 6.179(0.256) | 0.871(0.104) |
| GGL | 0.966(0.010) | 0.769(0.043) | 0.552(0.054) | 0.879(0.022) | 5.274(0.122) | 0.529(0.029) |
| | | | $n = 100$, $p = 100$, $\rho = 0.5$ | | | |
| Our method ($\alpha = 1$) | 0.992(0.002) | 0.664(0.043) | 0.699(0.023) | **0.920(0.030)** | 3.170(0.170) | **0.426(0.050)** |
| Our method ($\alpha = n$) | 0.986(0.008) | 0.770(0.043) | **0.713(0.035)** | 0.882(0.020) | 3.256(0.112) | 0.427(0.043) |
| BAGUS | 0.986(0.002) | 0.710(0.030) | 0.667(0.023) | 0.878(0.014) | 3.707(0.146) | 0.587(0.089) |
| Pooled | 0.976(0.003) | 0.469(0.031) | 0.421(0.027) | 0.777(0.033) | 5.538(0.208) | 0.735(0.111) |
| GGL | 0.980(0.007) | 0.684(0.077) | 0.608(0.028) | 0.838(0.038) | 4.940(0.256) | 0.502(0.026) |

Table 2: Result of scale-free network

| | Spec | Sens | MCC | AUC | F-norm | $\ell_\infty$ norm |
|---|---|---|---|---|---|---|
| | | | $n = 100$, $p = 100$, $\rho = 0$ | | | |
| Our method ($\alpha = 1$) | 1.000(0.000) | 1.000(0.002) | **0.993(0.006)** | **1.000(0.000)** | **1.664(0.088)** | 0.514(0.117) |
| Our method ($\alpha = n$) | 0.996(0.002) | 0.976(0.014) | 0.906(0.047) | 0.988(0.007) | 1.942(0.133) | **0.432(0.092)** |
| BAGUS | 0.997(0.001) | 0.995(0.004) | 0.936(0.019) | 0.998(0.002) | 1.747(0.096) | 0.492(0.107) |
| Pooled | 0.958(0.003) | 0.746(0.043) | 0.429(0.027) | 0.903(0.018) | 7.148(0.300) | 0.869(0.024) |
| GGL | 0.938(0.007) | 1.000(0.001) | 0.483(0.022) | 1.000(0.001) | 5.043(0.282) | 0.545(0.019) |
| | | | $n = 100$, $p = 100$, $\rho = 0.25$ | | | |
| Our method ($\alpha = 1$) | 0.993(0.001) | 0.921(0.024) | **0.833(0.011)** | **0.992(0.003)** | **2.032(0.079)** | 0.454(0.087) |
| Our method ($\alpha = n$) | 0.991(0.002) | 0.914(0.020) | 0.808(0.021) | 0.955(0.010) | 2.365(0.083) | **0.435(0.050)** |
| BAGUS | 0.990(0.001) | 0.919(0.021) | 0.801(0.019) | 0.967(0.009) | 2.407(0.100) | 0.518(0.088) |
| Pooled | 0.959(0.004) | 0.654(0.040) | 0.415(0.027) | 0.833(0.021) | 6.331(0.229) | 0.799(0.040) |
| GGL | 0.959(0.006) | 0.964(0.013) | 0.591(0.029) | 0.980(0.007) | 4.705(0.137) | 0.540(0.024) |
| | | | $n = 100$, $p = 100$, $\rho = 0.5$ | | | |
| Our method ($\alpha = 1$) | 0.988(0.001) | 0.787(0.031) | **0.719(0.018)** | **0.958(0.009)** | **2.548(0.088)** | **0.437(0.058)** |
| Our method ($\alpha = n$) | 0.983(0.007) | 0.816(0.042) | 0.696(0.036) | 0.904(0.020) | 2.813(0.099) | 0.443(0.046) |
| BAGUS | 0.986(0.003) | 0.822(0.023) | 0.716(0.028) | 0.938(0.011) | 3.106(0.131) | 0.595(0.116) |
| Pooled | 0.972(0.003) | 0.508(0.032) | 0.405(0.025) | 0.761(0.030) | 5.816(0.178) | 0.741(0.052) |
| GGL | 0.978(0.007) | 0.847(0.041) | 0.672(0.035) | 0.921(0.021) | 4.808(0.275) | 0.539(0.030) |

Table 3: Average computational time (in seconds) based on 10 replications.

| | Nearest-neighbor Network | | | Scale-free Network | | |
|---|---|---|---|---|---|---|
| | $\rho = 0$ | $\rho = 0.25$ | $\rho = 0.5$ | $\rho = 0$ | $\rho = 0.25$ | $\rho = 0.5$ |
| Our method ($\alpha = 1$) | 3.667(0.040) | 3.645(0.087) | 3.552(0.026) | 3.556(0.037) | 3.545(0.030) | 3.537(0.033) |
| Our method ($\alpha = n$) | 7.792(0.456) | 4.596(0.643) | 3.597(0.049) | 5.285(2.623) | 3.600(0.025) | 3.578(0.023) |
| BAGUS | 3.635(0.023) | 3.572(0.027) | 3.547(0.021) | 3.553(0.012) | 3.546(0.022) | 3.534(0.018) |
| Pooled | 1.211(0.010) | 1.178(0.013) | 1.169(0.008) | 1.184(0.015) | 1.173(0.008) | 1.168(0.010) |
| GGL | 8.715(0.314) | 8.034(0.689) | 5.482(1.528) | 8.086(0.262) | 6.139(0.678) | 3.074(0.270) |

We divide the observations into $K = 3$ classes by year as it is natural to expect the precision matrix changes over year due to annual policy decisions, economic conditions, and other aspects of the business. Then, we take the first $80\%$ of observations in each class as training data and the other $20\%$ as test data.

We apply our method with $\alpha = 365$ as well as other methods we compared in the simulation studies, i.e., BAGUS, Pooled, and GGL, on the training data to estimate $\mu_k$'s and $\Theta_k$'s. For year $k$ and day $i$, we divide the data $Y_{k,i} = (y_{k,i}^{(1)}, \ldots, y_{k,i}^{(237)})$ into two parts, $Y_{k,i_1} = (y_{k,i}^{(1)}, \ldots, y_{k,i}^{(118)})$ and $Y_{k,i_2} = (y_{k,i}^{(119)}, \ldots, y_{k,i}^{(237)})$. Assuming the first half $Y_{k,i_1}$ is observed, we predict the second half $Y_{k,i_2}$ by the following best linear predictor derived from the multivariate Gaussian distribution:

$$\hat{Y}_{k,i_2} = \mathbb{E}(Y_{k,i_2} \mid Y_{k,i_1}) = \hat{\mu}_{k_2} + \hat{\Theta}_{k_{21}} \hat{\Theta}_{k_{11}}^{-1}(Y_{k,i_1} - \hat{\mu}_{k_1}), \text{ for } k = 1, 2, 3, \text{ and } i \in \mathcal{T}_k,$$

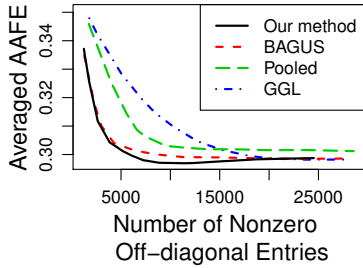

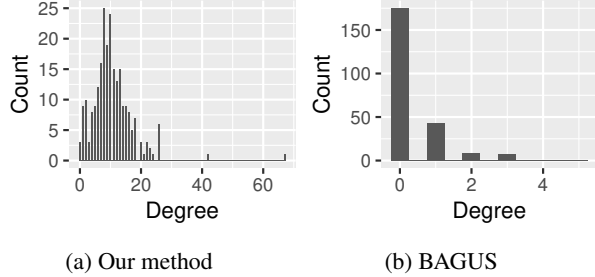

(a) Our method        (b) BAGUS

Figure 1: Averaged AAFE versus the total number of nonzero off-diagonal entries in the estimated precision matrices.

Figure 2: Degree distributions of the estimated common station networks over three years by our method and BAGUS.

where $\mathcal{T}_k$ is the index set of the $k$-th class of test data, $\mu_k = (\mu_{k_1}, \mu_{k_2})$ and $\Theta_k = \begin{pmatrix} \Theta_{k_{11}} & \Theta_{k_{12}} \\ \Theta_{k_{21}} & \Theta_{k_{22}} \end{pmatrix}$.

We use the average absolute forecast error (AAFE) of each class for performance comparison:

$$\text{AAFE}_k = \frac{1}{119} \sum_{j=119}^{237} \frac{1}{card(\mathcal{T}_k)} \sum_{i \in \mathcal{T}_k} |\hat{y}_{k,i}^{(j)} - y_{k,i}^{(j)}|, \ k = 1, 2, 3.$$

In Figure 1, we plot the averaged AAFE versus the number of nonzero off-diagonal entries in the estimated precision matrices. For our method, BAGUS, and Pooled methods, we plot the curves by fixing $v_1$ and varying $v_0$. For GGL, we fix the ratio between its two tuning parameters and varying them together. Different ratios would output similar curves and only one of them is plotted. We observe that our method not only achieves the lowest averaged AAFE, but also outputs the sparsest estimated precision matrices when the lowest averaged AAFE is attained.

To get estimates for the station networks, we select the hyperparameters of our method and BAGUS by BIC and and plot the degree distributions of the estimated common station networks over three years in Figure 2. From the common structure learned by our method, two stations are found to with higher connectivity and identified as hubs. It turns out that one is close to Union Station (a major transportation hub) and the other is close to Dupont Circle (a popular residential neighborhood). Therefore, it is not surprising the two stations play an important role in the dependence graph.

## Acknowledgment

This work is supported in part by grants NSF DMS-1916472 and NSF DMS-1811768.

## Footnotes

[1] Data available at `https://www.capitalbikeshare.com/system-data`

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
