[Supplementary Material]

# Supplementary Material for Bayesian Joint Estimation of Multiple Graphical Models

## 1 Proofs of the Main Results

In this section, we prove main results in the main paper. For simplicity, we assume sample sizes of the $K$ classes are the same: $n_1 = \cdots = n_K = n$.

Recall our objective function is

$$L(\boldsymbol{\Theta}) = \sum_{k=1}^{K} -\ell(\Theta_k) + \alpha \sum_{i=1}^{p} \sum_{k=1}^{K} \mathrm{pen}_1(\theta_{k,ii}) + \frac{\alpha}{2} \sum_{i \neq j} \mathrm{pen}_2(\boldsymbol{\theta}_{ij}), \tag{1.1}$$

where

$$\ell(\Theta_k) = -\frac{n}{2} \left( \mathrm{tr}(S_k \Theta_k) - \log \det(\Theta_k) \right),$$

$$\mathrm{pen}_1(\theta_{k,ii}) = \tau \theta_{k,ii},$$

$$\mathrm{pen}_2(\boldsymbol{\theta}_{ij}) = -\log \left( \frac{p_1}{(2v_1)^K} e^{-\|\boldsymbol{\theta}_{ij}\|_1/v_1} + \frac{1-p_1}{(2v_0)^K} e^{-\|\boldsymbol{\theta}_{ij}\|_1/v_0} \right) + C,$$

for a constant $C$ that makes $\mathrm{pen}_2(\mathbf{0}) = 0$.

### 1.1 Proof of Theorem 1

In this subsection, we show $L(\Theta)$ is strictly convex, under the constraints that $\Theta_k > 0$, $\|\Theta_k\|_2 \leqslant B$, $k = 1, \ldots, K$.

Decompose $L(\boldsymbol{\Theta})$ into the following two parts $L_1(\boldsymbol{\Theta}) + L_2(\boldsymbol{\Theta})$:

$$L_1(\boldsymbol{\Theta}) = \sum_k L_1(\Theta_k) = \sum_k \left[ -\ell(\Theta_k) - \frac{\alpha K}{8v_0^2} \|\Theta_k\|_F^2 \right],$$

$$L_2(\boldsymbol{\Theta}) = \alpha \sum_i \sum_k \mathrm{pen}_1(\theta_{k,ii}) + \frac{\alpha}{2} \sum_{i \neq j} \mathrm{pen}_2(\boldsymbol{\theta}_{ij}) + \frac{\alpha K}{8v_0^2} \sum_k \|\Theta_k\|_F^2.$$

We show $L_1(\boldsymbol{\Theta})$ and $L_2(\boldsymbol{\Theta})$ are both strictly convex, so that $L(\boldsymbol{\Theta})$ is strictly convex.

For $L_1(\boldsymbol{\Theta}) = \sum_k L_1(\Theta_k)$, the Hessian matrix of $L_1(\Theta_k)$ is as follows:

$$\nabla^2 L_1(\Theta_k) = \frac{n}{2} (\Theta_k \otimes \Theta_k)^{-1} - \frac{\alpha K}{4v_0^2} I_{p^2 \times p^2}.$$

The minimum eigenvalue of the Hessian matrix above can be bounded by

$$\lambda_{\min}(\nabla^2 L_1(\Theta_k)) = \frac{n}{2} \lambda_{\max}^{-1}(\Theta_k \otimes \Theta_k) - \frac{\alpha K}{4v_0^2}$$

$$= \frac{n}{2}\lambda_{\max}^{-2}(\Theta_k) - \frac{\alpha K}{4v_0^2}$$

$$\geqslant \frac{n}{2}\|\Theta_k\|_2^{-2} - \frac{\alpha K}{4v_0^2}$$

$$\geqslant \frac{n}{2}B^{-2} - \frac{\alpha K}{4v_0^2} > 0.$$

Therefore, $L_1(\Theta_k), k = 1, \ldots, K$ and $L_1(\boldsymbol{\Theta}) = \sum_k L_1(\Theta_k)$ are all strictly convex.

For $L_2(\boldsymbol{\Theta})$, let $\text{pen}(\boldsymbol{\Theta}) = \sum_i \sum_k \text{pen}_1(\theta_{k,ii}) + \frac{1}{2}\sum_{i \neq j}\text{pen}_2(\boldsymbol{\theta}_{ij})$. We now quantify the magnitude of its second-order subgradients. For any $k, k' \in \{1, ..., K\}$ and $i \neq j$, we have

$$\begin{cases} \left|\nabla^2_{\theta_{k,ij},\theta_{k',ij}}\text{pen}(\boldsymbol{\Theta})\right| = (\frac{1}{v_0} - \frac{1}{v_1})^2 \dfrac{\frac{p_1}{1-p_1}(\frac{v_0}{v_1})^K e^{(\frac{1}{v_0} - \frac{1}{v_1})\|\boldsymbol{\theta}_{ij}\|_1}}{\left[1 + \frac{p_1}{1-p_1}(\frac{v_0}{v_1})^K e^{(\frac{1}{v_0} - \frac{1}{v_1})\|\boldsymbol{\theta}_{ij}\|_1}\right]^2} \leqslant \dfrac{1}{4v_0^2} \\[2em] \nabla^2_{\theta_{k,ii},\theta_{k',ii}}\text{pen}(\boldsymbol{\Theta}) = 0. \end{cases}$$

Therefore, for any $(i, j)$ pair, we have

$$\lambda_{\min}(\nabla^2_{\boldsymbol{\theta}_{ij},\boldsymbol{\theta}_{ij}}\text{pen}(\boldsymbol{\Theta})) > -\frac{K}{4v_0^2}.$$

Hence, $\nabla^2_{\boldsymbol{\theta}_{ij},\boldsymbol{\theta}_{ij}}L_2(\boldsymbol{\Theta})$ is positive definite for any $(i, j)$ pair considering

$$\lambda_{\min}(\nabla^2_{\boldsymbol{\theta}_{ij},\boldsymbol{\theta}_{ij}}L_2(\boldsymbol{\Theta})) = \alpha\lambda_{\min}(\nabla^2_{\boldsymbol{\theta}_{ij},\boldsymbol{\theta}_{ij}}\text{pen}(\boldsymbol{\Theta})) + \frac{\alpha K}{4v_0^2}$$

$$> -\frac{\alpha K}{4v_0^2} + \frac{\alpha K}{4v_0^2} = 0,$$

Since $\nabla^2_{\boldsymbol{\theta}_{ij},\boldsymbol{\theta}_{i'j'}}L_2(\boldsymbol{\Theta}) = \mathbf{0}$ for any different pairs $(i, j) \neq (i', j')$, we have $\nabla^2 L_2(\boldsymbol{\Theta})$ is positive definite. Therefore, $L_2(\boldsymbol{\Theta})$ is strictly convex.

## 1.2 Proof of Theorem 2

In this subsection, we provide the proof of Theorem 2 of estimation consistency in the main paper.

We first introduce some notations that we use in the proof. Let $\Sigma_k^0$ denote the $k$'th true covariance matrix and $W_k = S_k - \Sigma_k^0$ denote the difference between the $k$'th sample covariance matrix and $\Sigma_k^0$. Use $\boldsymbol{\Sigma}^0 = (\Sigma_1^0, \ldots, \Sigma_K^0)$ and $\boldsymbol{W} = (W_1, \ldots, W_K)$ to denote the collection of $\Sigma_k^0$'s and $W_k$'s, respectively. For any subset $\mathcal{M}$ of $\{(i, j) : 1 \leqslant i, j \leqslant p\}$ and a $p \times p$ matrix $\Theta_k$, let $(\Theta_k)_{\mathcal{M}}$ denote the submatrix of $\Theta_k$ with entries indexed by $\mathcal{M}$; for $\boldsymbol{\Theta} = (\Theta_1, \ldots, \Theta_K)$, let $\boldsymbol{\Theta}_{\mathcal{M}} = ((\Theta_1)_{\mathcal{M}}, \ldots, (\Theta_K)_{\mathcal{M}})$. We use $\Gamma^0_{\mathcal{M}\mathcal{M}}$ to denote the Hessian matrix $\nabla^2_{\boldsymbol{\Theta}_{\mathcal{M}},\boldsymbol{\Theta}_{\mathcal{M}}}(-\sum_k \log \det \Theta_k)$ evaluated at $\boldsymbol{\Theta}^0$, and $(\Gamma_k^0)_{\mathcal{M}\mathcal{M}}$ to denote the Hessian matrix $\nabla^2_{(\Theta_k)_{\mathcal{M}},(\Theta_k)_{\mathcal{M}}}(-\log \det \Theta_k)$ evaluated at $\Theta_k^0$.

Our proof is motivated by the proof techniques from [6] and [2] and we prove Theorem 2 through three steps:

- Step 1. Construct a solution set $\mathcal{A}$ for the following constrained minimization problem:

$$\min_{\boldsymbol{\Theta}\in\Omega_1} L(\boldsymbol{\Theta}),$$

  where $\Omega_1 := \Omega \cap \{\boldsymbol{\Theta} : \boldsymbol{\Theta}_{\mathcal{B}^c} = 0\}$ with $\mathcal{B} = \{(i, j) : i = j \text{ or } |\theta_{k,ij}^0| \geqslant C_5\sqrt{\log p/n} \text{ for some } k\}$ and $\Omega = \{\boldsymbol{\Theta} : \Theta_k > 0, \|\Theta_k\|_2 \leqslant B, k = 1, \ldots, K\}$ is the original constraint space.

- Step 2. Show that there exists $\tilde{\boldsymbol{\Theta}} \in \mathcal{A}$ such that $\|\tilde{\boldsymbol{\Theta}} - \boldsymbol{\Theta}^0\|_{\infty} < r_e = C_5\sqrt{\frac{\log p}{n}}$.

- Step 3. Show that the constructed $\tilde{\boldsymbol{\Theta}}$ in Step 2 is a local minimizer of $L(\boldsymbol{\Theta})$. Then, from the convexity of the constrained minimization problem, $\tilde{\boldsymbol{\Theta}}$ must be the global minimizer.

Let $M_{\Gamma^0} = |||(\Gamma^0_{\mathcal{BB}})^{-1}|||_\infty = \sup_k |||(\Gamma^0_k)_{\mathcal{BB}}^{-1}|||_\infty$ and $M_{\Sigma^0} = \sup_k |||\Sigma^0_k|||_\infty$ where $||| \cdot |||_\infty$ denotes the matrix maximum absolute row sum norm. We first present a more general theorem below. Theorem 2 in the main paper directly follows from the following theorem by checking its condition $(v)$ with tail conditions and standard concentration inequalities (c.f., (the proof of) Theorem 1 of [1]).

**Theorem 4** *Under the following conditions:*

*(i) rates of the hyperparameters $v_1$, $v_0$, $p_1$, and $\tau$:*

$$\frac{\alpha}{nv_1} < C_3/(1 + \epsilon_1)\sqrt{\log p/n},$$

$$\frac{\alpha}{nv_0} > C_4\sqrt{\log p/n},$$

$$\epsilon_2 < \frac{v_1^K(1 - p_1)}{v_0^K p_1} < \frac{v_1^{K+2}(1 - p_1)}{v_0^{K+2} p_1} \leqslant \epsilon_1 p^{\epsilon_0/\alpha},$$

$$\frac{\alpha\tau}{n} \leqslant \frac{C_3}{2}\sqrt{\log p/n},$$

*where $\epsilon_0 = (C_2 - C_3)M_{\Gamma^0}(C_4 - C_3)$ with $C_2 > C_3 > 0$, $C_4 > (1 + \epsilon_2)/\epsilon_2[C_1 + 2(C_1 + C_2)M_{\Gamma^0}M^2_{\Sigma^0} + 6(C_1 + C_2)^2dM^2_{\Gamma^0}M^3_{\Sigma^0}/M]$, and $\epsilon_1 > \epsilon_2 > 0$,*

*(ii) the eigenvalues of the true precision matrices:*

$$1/\xi_0 < \lambda_{\min}(\Theta^0_k) < \lambda_{\max}(\Theta^0_k) \leqslant 1/\xi_1 < \sqrt{\frac{2\alpha p^{\epsilon_0/\alpha}}{KC_3^2\log p}} \quad \text{for } k = 1, \ldots, K,$$

*(iii) the sample size $n$: $\sqrt{n} \geqslant M\sqrt{\log p}$ with*

$$M = 2d(C_1 + C_2)M_{\Gamma^0}\max\{2M_{\Gamma^0}(2M_{\Sigma^0}(M^2_{\Sigma^0} + \frac{3}{2}dM^3_{\Sigma^0}) + (M^2_{\Sigma^0} + \frac{3}{2}M^3_{\Sigma^0})^2), 1, 2/\xi_1^2, \xi_0\},$$

*(iv) the bounds on the spectral norms of the estimated precision matrices:*

$$1/\xi_1 + 2d(C_1 + C_2)M_{\Gamma^0}\sqrt{\log p/n} < B < \left(\frac{2nv_0^2}{\alpha K}\right)^{1/2},$$

*(v) difference between the sample covariance matrices and the true covariance matrices: $\|W\|_\infty \leqslant C_1\sqrt{\log p/n}$,*

*we have*

$$\|\hat{\Theta} - \Theta^0\|_\infty < 2(C_1 + C_2)M_{\Gamma^0}\sqrt{\log p/n},$$

*where $C_2$ is chosen such that $2(C_1 + C_2)M_{\Gamma^0} = C_5$.*

**Proof of Theorem 4**

- Step 1. Construct a solution set for minimizing the objective function (1.1) as follows:

$$\mathcal{A} = \{\Theta : 0 \in G(\Theta)_{\mathcal{B}}, \Theta_{\mathcal{B}^c} = 0\} \cap \Omega,$$

where $\mathcal{B} = \{(i, j) : i = j \text{ or } |\theta^0_{k,ij}| \geqslant r_e \text{ for some } k\}$ for $r_e = 2(C_1 + C_2)M_{\Gamma^0}\sqrt{\log p/n}$ and $G(\Theta) = \frac{n}{2}(S - \Theta^{-1} + \frac{2\alpha}{n}Z(\Theta))$ is the subgradient of the objective function $L(\Theta)$ with

$$Z_{k,ij}(\Theta) = Z_{k,ij}(\theta_{ij}) = \begin{cases} \tau & \text{if } i = j \\ \frac{1}{2}\frac{\partial}{\partial|\theta_{k,ij}|}\text{pen}_2(\theta_{ij})\text{sign}(\theta_{k,ij}) & \text{if } i \neq j \end{cases}$$

where

$$\frac{\partial}{\partial|\theta_{k,ij}|}\text{pen}_2(\theta_{ij}) = w(\theta_{ij})\frac{1}{v_1} + (1 - w(\theta_{ij}))\frac{1}{v_0},$$

and

$$\text{sign}(\theta_{k,ij}) = \begin{cases} 1 & \text{when } \theta_{k,ij} > 0, \\ -1 & \text{when } \theta_{k,ij} < 0, \\ [-1, 1] & \text{when } \theta_{k,ij} = 0. \end{cases}$$

- Step 2. Show that there exists some $\tilde{\boldsymbol{\Theta}} \in \mathcal{A}$ such that $\|\tilde{\boldsymbol{\Theta}} - \boldsymbol{\Theta}^0\|_\infty < r_e$.

  We only need to show for the entries of $\boldsymbol{\Theta}$ indexed by $\mathcal{B}$, because $\|(\tilde{\boldsymbol{\Theta}} - \boldsymbol{\Theta}^0)_{\mathcal{B}^c}\|_\infty < r_e$ by the way that $\mathcal{A}$ is constructed.

  Define the following mapping from $\mathbb{R}^{K|\mathcal{B}|}$ to $\mathbb{R}^{K|\mathcal{B}|}$:

  $$F(\mathrm{vec}(\boldsymbol{\Delta}_{\mathcal{B}})) = -\frac{2}{n}(\Gamma_{\mathcal{B}\mathcal{B}}^0)^{-1}\mathrm{vec}\left(\tilde{G}(\boldsymbol{\Theta}^0 + \boldsymbol{\Delta})_{\mathcal{B}}\right) + \mathrm{vec}(\boldsymbol{\Delta}_{\mathcal{B}}),$$

  where $\tilde{G}(\boldsymbol{\Theta}) = \frac{n}{2}(\boldsymbol{S} - \boldsymbol{\Theta}^{-1} + \frac{2\alpha}{n}\tilde{\boldsymbol{Z}})$ with $\tilde{\boldsymbol{Z}}$ to be a member of the subgradient of the unique minimizer of (1.1) under the constraint set $\Omega$, and $\boldsymbol{\Delta}$ satisfies $(\boldsymbol{\Theta}^0 + \boldsymbol{\Delta})_{\mathcal{B}^c} = \boldsymbol{0}$, that is, $\boldsymbol{\Delta}_{\mathcal{B}^c} = -\boldsymbol{\Theta}_{\mathcal{B}^c}^0$. We will first show $F(\mathbb{B}(r)) \subseteq \mathbb{B}(r)$ for the $\ell_\infty$ ball $\mathbb{B}(r)$ in $\mathbb{R}^{K|\mathcal{B}|}$ with $r = 2(C_1 + C_3)M_{\Gamma^0}\sqrt{\log p/n} < r_e$. Then, by Brouwer's fixed point theorem [4], there exists a fixed point $\mathrm{vec}(\tilde{\boldsymbol{\Delta}}_{\mathcal{B}}) \in \mathbb{B}(r)$ such that $F(\mathrm{vec}(\tilde{\boldsymbol{\Delta}}_{\mathcal{B}})) = \mathrm{vec}(\tilde{\boldsymbol{\Delta}}_{\mathcal{B}})$. Equivalently, we have a fixed point $\mathrm{vec}(\tilde{\boldsymbol{\Delta}}_{\mathcal{B}}) \in \mathbb{B}(r)$ such that $\tilde{G}(\boldsymbol{\Theta}^0 + \tilde{\boldsymbol{\Delta}})_{\mathcal{B}} = \boldsymbol{0}$ or $\boldsymbol{0} \in G(\boldsymbol{\Theta}^0 + \tilde{\boldsymbol{\Delta}})_{\mathcal{B}}$.

  For any $\mathrm{vec}(\boldsymbol{\Delta}_{\mathcal{B}}) \in \mathbb{B}(r)$, we have

  $$
  \begin{aligned}
  F(\mathrm{vec}(\boldsymbol{\Delta}_{\mathcal{B}})) = & -(\Gamma_{\mathcal{B}\mathcal{B}}^0)^{-1}\mathrm{vec}\left(\left(\boldsymbol{S} - (\boldsymbol{\Theta}^0 + \boldsymbol{\Delta})^{-1} + \frac{2\alpha}{n}\tilde{\boldsymbol{Z}}\right)_{\mathcal{B}}\right) + \mathrm{vec}(\boldsymbol{\Delta}_{\mathcal{B}}) \\
  = & -(\Gamma_{\mathcal{B}\mathcal{B}}^0)^{-1}\left(\mathrm{vec}\left((\boldsymbol{\Sigma}^0 - (\boldsymbol{\Theta}^0 + \boldsymbol{\Delta})^{-1})_{\mathcal{B}}\right) - (\Gamma_{\mathcal{B}\mathcal{B}}^0)\mathrm{vec}(\boldsymbol{\Delta}_{\mathcal{B}})\right) \quad (1.2) \\
  & -(\Gamma_{\mathcal{B}\mathcal{B}}^0)^{-1}\mathrm{vec}\left(\boldsymbol{W}_{\mathcal{B}} + \frac{2\alpha}{n}\tilde{\boldsymbol{Z}}_{\mathcal{B}}\right) \quad (1.3)
  \end{aligned}
  $$

  where $(\boldsymbol{\Theta}^0 + \boldsymbol{\Delta})^{-1} = ((\Theta_1^0 + \Delta_1)^{-1}, \ldots, (\Theta_K^0 + \Delta_K)^{-1})$.

  For (1.2), we have

  $$
  \begin{aligned}
  & \left\|-(\Gamma_{\mathcal{B}\mathcal{B}}^0)^{-1}\left(\mathrm{vec}\left((\boldsymbol{\Sigma}^0 - (\boldsymbol{\Theta}^0 + \boldsymbol{\Delta})^{-1})_{\mathcal{B}}\right) - (\Gamma_{\mathcal{B}\mathcal{B}}^0)\mathrm{vec}(\boldsymbol{\Delta}_{\mathcal{B}})\right)\right\|_\infty \\
  \leqslant & \sup_k \left\|-(\Gamma_k^0)_{\mathcal{B}\mathcal{B}}^{-1}\left(\mathrm{vec}\left((\Sigma_k^0 - (\Theta_k^0 + \Delta_k)^{-1})_{\mathcal{B}}\right) - (\Gamma_k^0)_{\mathcal{B}\mathcal{B}}\mathrm{vec}((\Delta_k)_{\mathcal{B}})\right)\right\|_\infty \\
  \leqslant & \sup_k \||(\Gamma_k^0)_{\mathcal{B}\mathcal{B}}^{-1}\||_\infty \left\|\mathrm{vec}\left((\Sigma_k^0 - (\Theta_k^0 + \Delta_k)^{-1})_{\mathcal{B}}\right) - (\Gamma_k^0)_{\mathcal{B}\mathcal{B}}\mathrm{vec}((\Delta_k)_{\mathcal{B}})\right\|_\infty \\
  \leqslant & M_{\Gamma^0} \sup_k \left\|\int_0^1 \left\{(\Theta_k^0 + t\Delta_k)^{-1} \otimes (\Theta_k^0 + t\Delta_k)^{-1} - (\Theta_k^0)^{-1} \otimes (\Theta_k^0)^{-1}\right\} dt\right\|_\infty \|\mathrm{vec}((\Delta_k)_{\mathcal{B}})\|_\infty \\
  \leqslant & M_{\Gamma^0} r \sup_k \int_0^1 \||(\Theta_k^0 + t\Delta_k)^{-1} \otimes (\Theta_k^0 + t\Delta_k)^{-1} - (\Theta_k^0)^{-1} \otimes (\Theta_k^0)^{-1}\||_\infty dt \\
  \leqslant & M_{\Gamma^0}(2M_{\boldsymbol{\Sigma}^0}(M_{\boldsymbol{\Sigma}^0}^2 + \frac{3}{2}dM_{\boldsymbol{\Sigma}^0}^3)dr + (M_{\boldsymbol{\Sigma}^0}^2 + \frac{3}{2}M_{\boldsymbol{\Sigma}^0}^3)^2 d^2 r^2)r \\
  \leqslant & \frac{r}{2},
  \end{aligned}
  $$

  $$(1.4)$$

  where the second inequality is because $\|AB\|_\infty \leqslant \||A\||_\infty \|B\|_\infty$, the third inequality is because $\Gamma_k^0 = (\Theta_k^0)^{-1} \otimes (\Theta_k^0)^{-1}$, the fifth inequality is because $\|(\Theta_k^0 + \Delta_k)^{-1} - (\Theta_k^0)^{-1}\|_\infty \leqslant (M_\Sigma^2 + \frac{3}{2}M_\Sigma^3)r$ from Corollary 4 of [5], $\||(\Theta_k^0 + \Delta_k)^{-1} - (\Theta_k^0)^{-1}\||_\infty \leqslant d\|(\Theta_k^0 + \Delta_k)^{-1} - (\Theta_k^0)^{-1}\|_\infty$ and Lemma 13 of [3], and last inequality is by $M_{\Gamma^0}(2M_{\boldsymbol{\Sigma}^0}(M_{\boldsymbol{\Sigma}^0}^2 + \frac{3}{2}dM_{\boldsymbol{\Sigma}^0}^3) + (M_{\boldsymbol{\Sigma}^0}^2 + \frac{3}{2}M_{\boldsymbol{\Sigma}^0}^3)^2)dr < \frac{1}{2}$ and $r < \frac{1}{d}$ from condition (iii).

  For (1.3), we have

  $$
  \begin{aligned}
  \left\|(\Gamma_{\mathcal{B}\mathcal{B}}^0)^{-1}\mathrm{vec}\left(\boldsymbol{W}_{\mathcal{B}} + \frac{2\alpha}{n}\tilde{\boldsymbol{Z}}_{\mathcal{B}}\right)\right\|_\infty & \leqslant \sup_k \left\|((\Gamma_k^0)_{\mathcal{B}\mathcal{B}})^{-1}\mathrm{vec}\left((W_k)_{\mathcal{B}} + \frac{2\alpha}{n}(\tilde{Z}_k)_{\mathcal{B}}\right)\right\|_\infty \\
  & \leqslant \sup_k M_{\Gamma^0}\left(\|W_k\|_\infty + \left\|\frac{2\alpha}{n}(\tilde{Z}_k)_{\mathcal{B}}\right\|_\infty\right) \\
  & \leqslant M_{\Gamma^0}\left(C_1\sqrt{\log p/n} + C_3\sqrt{\log p/n}\right) = \frac{r}{2}, \quad (1.5)
  \end{aligned}
  $$

  where the third inequality is because of the upper bound on the magnitude of the first derivatives from Lemma 5 and $\|W_k\|_\infty \leqslant \|\boldsymbol{W}\|_\infty \leqslant C_1\sqrt{\log p/n}$ from condition (v).

Thus, combining (1.3), (1.4), and (1.5), we have $\|F(\text{vec}(\boldsymbol{\Delta}_{\mathcal{B}}))\|_\infty \leqslant r$, that is, $F(\text{vec}(\boldsymbol{\Delta}_{\mathcal{B}})) \subseteq \mathbb{B}(r)$ for any $\text{vec}(\boldsymbol{\Delta}_{\mathcal{B}}) \in \mathbb{B}(r)$. Therefore, we have $F(\mathbb{B}(r)) \subseteq \mathbb{B}(r)$. By Brouwer's fixed point theorem [4], there exists a fixed point $\text{vec}(\tilde{\boldsymbol{\Delta}}_{\mathcal{B}}) \in \mathbb{B}(r)$ such that $F(\text{vec}(\tilde{\boldsymbol{\Delta}}_{\mathcal{B}})) = \text{vec}(\tilde{\boldsymbol{\Delta}}_{\mathcal{B}})$, which is equivalent to $\tilde{G}(\boldsymbol{\Theta}^0 + \tilde{\boldsymbol{\Delta}})_{\mathcal{B}} = \mathbf{0}$ or $\mathbf{0} \in G(\boldsymbol{\Theta}^0 + \tilde{\boldsymbol{\Delta}})_{\mathcal{B}}$. Let $\tilde{\boldsymbol{\Theta}} = \boldsymbol{\Theta}^0 + \tilde{\boldsymbol{\Delta}}$, we have

$$
\begin{cases}
\tilde{\boldsymbol{\Theta}}_{\mathcal{B}} = \boldsymbol{\Theta}^0_{\mathcal{B}} + \tilde{\boldsymbol{\Delta}}_{\mathcal{B}} \\
\tilde{\boldsymbol{\Theta}}_{\mathcal{B}^c} = \boldsymbol{\Theta}^0_{\mathcal{B}^c} + \tilde{\boldsymbol{\Delta}}_{\mathcal{B}^c} = \mathbf{0}.
\end{cases}
$$

In addition, we have this estimate $\tilde{\boldsymbol{\Theta}}$ satisfies $\mathbf{0} \in G(\tilde{\boldsymbol{\Theta}})_{\mathcal{B}}$, $\tilde{\boldsymbol{\Theta}}_{\mathcal{B}^c} = \mathbf{0}$, and $\|(\tilde{\boldsymbol{\Theta}} - \boldsymbol{\Theta}^0)_{\mathcal{B}}\|_\infty \leqslant r < r_e$. As long as $\tilde{\boldsymbol{\Theta}}$ is in the parameter space of interest $\Omega$, that is, $\|\tilde{\Theta}_k\|_2 < B$ and $\tilde{\Theta}_k > 0$, we have the statement for Step 2 as desired.

Indeed, for any $k$, by condition (iv), we have

$$
\begin{aligned}
\|\tilde{\Theta}_k\|_2 &\leqslant \|\tilde{\Theta}_k - \Theta^0_k\|_2 + \|\Theta^0_k\|_2 \\
&\leqslant \|\tilde{\Delta}_k\|_\infty + \frac{1}{\xi_1} \\
&\leqslant d r_e + \frac{1}{\xi_1} < B,
\end{aligned}
$$

and, by conditions (ii) and (iii), we have

$$
\lambda_{\min}(\tilde{\Theta}_k) \geqslant \lambda_{\min}(\Theta^0_k) - \|\tilde{\Delta}_k\|_2 \geqslant \lambda_{\min}(\Theta^0_k) - d r_e > 0.
$$

Therefore, we have $\tilde{\boldsymbol{\Theta}} \in \mathcal{A}$ and $\|\tilde{\boldsymbol{\Theta}} - \boldsymbol{\Theta}^0\|_\infty < r_e$.

- Step 3. Show that the constructed $\tilde{\boldsymbol{\Theta}}$ in Step 2 is a local minimizer of $L(\boldsymbol{\Theta})$.

  It suffices to show that there exists some $\epsilon > 0$ such that $H(\boldsymbol{\Delta}) = L(\tilde{\boldsymbol{\Theta}} + \boldsymbol{\Delta}) - L(\tilde{\boldsymbol{\Theta}}) \geqslant 0$ for any $\boldsymbol{\Delta}$ with $\|\boldsymbol{\Delta}\|_\infty < \epsilon$. We have

$$
\begin{aligned}
H(\boldsymbol{\Delta}) = \sum_{k=1}^{K} \frac{n}{2} &\left\{ \text{tr}(\Delta_k(S_k - \tilde{\Theta}_k^{-1})) + \text{tr}(\Delta_k \tilde{\Theta}_k^{-1}) - \left[ \log\det(\tilde{\Theta}_k + \Delta_k) - \log\det(\tilde{\Theta}_k) \right] \right\} \\
&+ \alpha \sum_{k=1}^{K} \sum_{i=1}^{p} \tau \Delta_{k,ii} + \frac{\alpha}{2} \sum_{i \neq j} \left[ \text{pen}_2(\tilde{\boldsymbol{\theta}}_{ij} + \boldsymbol{\Delta}_{ij}) - \text{pen}_2(\tilde{\boldsymbol{\theta}}_{ij}) \right].
\end{aligned}
$$

Under the condition $\sqrt{n} \geqslant 4(C_1 + C_2) M_{\Gamma^0} d \sqrt{\log p}/\xi_1^2$ and with the same proof for Theorem 1 in [7], we have the following upper bound on $\log\det(\tilde{\Theta}_k + \Delta_k) - \log\det(\tilde{\Theta}_k)$:

$$
\log\det(\tilde{\Theta}_k + \Delta_k) - \log\det(\tilde{\Theta}_k) \leqslant \text{tr}(\Delta_k \tilde{\Theta}_k^{-1}) - \frac{1}{4}\xi_1^2 \|\Delta_k\|_F^2, \quad k = 1, \ldots, K. \tag{1.6}
$$

For any $(i, j)$ pair in $\mathcal{B}^c$ with $i \neq j$, we have $\tilde{\boldsymbol{\theta}}_{ij} = \mathbf{0}$ and therefore, for some $u_1$ in $(0, 1)$,

$$
\text{pen}_2(\tilde{\boldsymbol{\theta}}_{ij} + \boldsymbol{\Delta}_{ij}) - \text{pen}_2(\tilde{\boldsymbol{\theta}}_{ij}) = \nabla_{\boldsymbol{\theta}_{ij}} \text{pen}_2(u_1 \boldsymbol{\Delta}_{ij}) \boldsymbol{\Delta}_{ij}. \tag{1.7}
$$

For any $(i, j)$ pair in $\mathcal{B}$ with $i \neq j$, for some $u_2$ in $(0, 1)$, we have

$$
\begin{aligned}
\text{pen}_2(\tilde{\boldsymbol{\theta}}_{ij} + \boldsymbol{\Delta}_{ij}) - \text{pen}_2(\tilde{\boldsymbol{\theta}}_{ij}) = &\sum_{k:\tilde{\theta}_{k,ij}=0} \frac{\partial}{\partial|\theta_{k,ij}|}\text{pen}_2(\tilde{\boldsymbol{\theta}}_{ij})|\Delta_{k,ij}| + \sum_{k:\tilde{\theta}_{k,ij}\neq 0} \frac{\partial}{\partial\theta_{k,ij}}\text{pen}_2(\tilde{\boldsymbol{\theta}}_{ij})\Delta_{k,ij} \\
&+ \frac{1}{2}\boldsymbol{\Delta}_{ij}^T \nabla^2_{\boldsymbol{\theta}_{ij},\boldsymbol{\theta}_{ij}}\text{pen}_2(\tilde{\boldsymbol{\theta}}_{ij} + u_2\boldsymbol{\Delta}_{ij})\boldsymbol{\Delta}_{ij}.
\end{aligned}
\tag{1.8}
$$

Combining (1.7) and (1.8), we have the following lower bound for $H(\boldsymbol{\Delta})$:

$$
\begin{aligned}
H(\boldsymbol{\Delta}) \geqslant &\sum_{k=1}^{K} \frac{n}{2} \left( \sum_{i,j} \Delta_{k,ij}(s_{k,ij} - \tilde{\Theta}_{k,ij}^{-1}) + \frac{1}{4}\xi_1^2\|\Delta_k\|_F^2 \right) + \alpha \sum_{k=1}^{K}\sum_{i=1}^{p}\tau\Delta_{k,ii} \\
&+ \frac{\alpha}{2} \sum_{i \neq j,\,(i,j)\in\mathcal{B}^c} \sum_{k=1}^{K} \frac{\partial}{\partial|\theta_{k,ij}|}\text{pen}_2(u_1\boldsymbol{\Delta}_{ij})|\Delta_{k,ij}|
\end{aligned}
$$

$$+ \frac{\alpha}{2} \sum_{i \neq j,\, (i,j) \in \mathcal{B}} \left[ \sum_{k:\tilde{\theta}_{k,ij}=0} \frac{\partial}{\partial |\theta_{k,ij}|} \mathrm{pen}_2(\tilde{\boldsymbol{\theta}}_{ij}) |\Delta_{k,ij}| + \sum_{k:\tilde{\theta}_{k,ij}\neq 0} \frac{\partial}{\partial \theta_{k,ij}} \mathrm{pen}_2(\tilde{\boldsymbol{\theta}}_{ij}) \Delta_{k,ij} \right.$$

$$\left. + \frac{1}{2} \boldsymbol{\Delta}_{ij}^T \nabla^2_{\boldsymbol{\theta}_{ij},\boldsymbol{\theta}_{ij}} \mathrm{pen}_2(\tilde{\boldsymbol{\theta}}_{ij} + u_2 \boldsymbol{\Delta}_{ij}) \boldsymbol{\Delta}_{ij} \right]$$

$$= \text{(I)} + \text{(II)} + \text{(III)},$$

where

$$\text{(I)} = \frac{n}{2} \sum_{k=1}^{K} \sum_{\tilde{\theta}_{k,ij} \neq 0, (i,j) \in \mathcal{B}} \Delta_{k,ij} \left( s_{k,ij} - \tilde{\Theta}_{k,ij}^{-1} + \frac{2\alpha}{n} Z_{k,ij}(\tilde{\boldsymbol{\theta}}_{ij}) \right)$$

$$+ \frac{n}{2} \sum_{k=1}^{K} \sum_{\tilde{\theta}_{k,ij} = 0, (i,j) \in \mathcal{B}} \left[ \Delta_{k,ij} \left( s_{k,ij} - \tilde{\Theta}_{k,ij}^{-1} \right) + |\Delta_{k,ij}| \frac{\alpha}{n} \frac{\partial}{\partial |\theta_{k,ij}|} \mathrm{pen}_2(\tilde{\boldsymbol{\theta}}_{ij}) \right],$$

$$\text{(II)} = \sum_{k=1}^{K} \frac{n}{8} \xi_1^2 \|\Delta_k\|_F^2 + \sum_{i \neq j,\, (i,j) \in \mathcal{B}} \frac{\alpha}{4} \boldsymbol{\Delta}_{ij}^T \nabla^2_{\boldsymbol{\theta}_{ij},\boldsymbol{\theta}_{ij}} \mathrm{pen}_2(\tilde{\boldsymbol{\theta}}_{ij} + u_2 \boldsymbol{\Delta}_{ij}) \boldsymbol{\Delta}_{ij},$$

$$\text{(III)} = \frac{n}{2} \sum_{i \neq j, (i,j) \in \mathcal{B}^c} \sum_{k=1}^{K} \left[ \Delta_{k,ij}(s_{k,ij} - \tilde{\Theta}_{k,ij}^{-1}) + |\Delta_{k,ij}| \frac{\alpha}{n} \frac{\partial}{\partial |\theta_{k,ij}|} \mathrm{pen}_2(u_1 \boldsymbol{\Delta}_{ij}) \right].$$

Due to the construction of $\tilde{\boldsymbol{\Theta}}$, we have (I) $\geqslant 0$.

For (II), by condition (ii) and the upper bound on the magnitude of the second derivatives of $\mathrm{pen}_2(\boldsymbol{\theta}_{ij})$ in Lemma 5, we have

$$\text{(II)} \geqslant \sum_{i,j} \sum_{k=1}^{K} \frac{n}{8} \xi_1^2 \Delta_{k,ij}^2 - \sum_{i \neq j,\, (i,j) \in \mathcal{B}} \frac{\frac{nC_3^2 \log p}{4\alpha p^{\epsilon_0/\alpha}}}{4} \left( \sum_{k=1}^{K} \Delta_{k,ij} \right)^2$$

$$\geqslant \sum_{i \neq j,\, (i,j) \in \mathcal{B}} n \left( \frac{1}{8K} \xi_1^2 - \frac{C_3^2 \log p}{16\alpha p^{\epsilon_0/\alpha}} \right) \left( \sum_{k=1}^{K} \Delta_{k,ij} \right)^2 \geqslant 0.$$

In (III), we have the following upper bound on the magnitude of $s_{k,ij} - \tilde{\Theta}_{k,ij}^{-1}$:

$$\left| s_{k,ij} - \tilde{\Theta}_{k,ij}^{-1} \right| \leqslant |s_{k,ij} - \sigma_{k,ij}^0| + |\tilde{\Theta}_{k,ij}^{-1} - \sigma_{k,ij}^0|$$

$$\leqslant C_1 \sqrt{\frac{\log p}{n}} + M_{\Sigma_k^0}^2 r_e + \frac{3}{2} d_k M_{\Sigma_k^0}^3 r_e^2,$$

where $|\tilde{\Theta}_{k,ij}^{-1} - \sigma_{k,ij}^0| < M_{\Sigma_k^0}^2 r_e + \frac{3}{2} d_k M_{\Sigma_k^0}^3 r_e^2$ is by Corollary 4 in [5].

For the rest of the first derivative inside the inner summation in (III), we have

$$\lim_{\boldsymbol{\Delta}_{ij} \to \boldsymbol{0}} \frac{\alpha}{n} \frac{\partial}{\partial |\theta_{k,ij}|} \mathrm{pen}_2(u_1 \boldsymbol{\Delta}_{ij}) = \frac{\alpha}{n} \left[ w(\boldsymbol{0}) \frac{1}{v_1} + (1 - w(\boldsymbol{0})) \frac{1}{v_0} \right]$$

$$> (1 - w(\boldsymbol{0})) \frac{\alpha}{n v_0}$$

$$> \frac{\epsilon_2}{1 + \epsilon_2} \frac{\alpha}{n v_0}$$

$$> C_1 \sqrt{\frac{\log p}{n}} + M_{\Sigma_k^0}^2 r_e + \frac{3}{2} d_k M_{\Sigma_k^0}^3 r_e^2$$

$$\geqslant \left| s_{k,ij} - \tilde{\Theta}_{k,ij}^{-1} \right|,$$

where the second inequality is due to $\epsilon_2 < v_1^K (1 - p_1)/(v_0^K p_1)$. Therefore, there exists some small enough $\epsilon > 0$ such that (III) $\geqslant 0$.

Hence, there exists some $\epsilon > 0$ such that $H(\boldsymbol{\Delta}) \geqslant$ (I) + (II) + (III) $\geqslant 0$. Due to the strict convexity of our problem, we conclude that the constructed $\tilde{\boldsymbol{\Theta}}$ is equal to the unique global solution $\hat{\boldsymbol{\Theta}}$.

■

## 1.3 Proof of Theorem 3

In this subsection, we provide the proof of Theorem 3 of selection consistency in the main paper.

- When $\boldsymbol{\theta}_{ij}^0 = \mathbf{0}$, we have the estimate $\hat{\boldsymbol{\theta}}_{ij} = \mathbf{0}$. Therefore,

$$w(\hat{\boldsymbol{\theta}}_{ij}) = w(\mathbf{0}) = \frac{1}{1 + \frac{1-p_1}{p_1}\left(\frac{v_1}{v_0}\right)^K} \leqslant t.$$

- On the other hand, when $\boldsymbol{\theta}_{ij}^0 \neq \mathbf{0}$, by the minimal signal strength assumption in Theorem 3, we have $\|\hat{\boldsymbol{\theta}}_{ij}\|_1 > (L_0 - C_5)\sqrt{\log p/n}$ with probability going to one. Thus,

$$w(\hat{\boldsymbol{\theta}}_{ij}) = \frac{1}{1 + \frac{1-p_1}{p_1}\left(\frac{v_1}{v_0}\right)^K \exp\left\{-\left(\frac{1}{v_0} - \frac{1}{v_1}\right)\|\hat{\boldsymbol{\theta}}_{ij}\|_1\right\}}$$

$$> \frac{1}{1 + \frac{1-p_1}{p_1}\left(\frac{v_1}{v_0}\right)^K p^{-(C_4-C_3)(L_0-C_5)/\alpha}}$$

$$> t.$$

Therefore, we have

$$\mathbb{P}(\hat{\mathcal{S}} = \mathcal{S}^0) \to 1.$$

## 2 Other Proof

The following lemma gives the upper bounds on the magnitudes of the first derivative and second derivative of $\text{pen}_2(\boldsymbol{\theta}_{ij})$ when $\boldsymbol{\theta}_{ij}$ is $O(\sqrt{\log p/n})$ away from the zero point $\mathbf{0}$.

**Lemma 5** *Under condition (i) in Theorem 4, if $\|\boldsymbol{\theta}_{ij}\|_1 \geqslant 2(C_2 - C_3)M_{\Gamma^0}\sqrt{\log p/n}$, then for any $k$ and $k'$, we have*

$$\frac{\alpha}{n}\left|\frac{\partial}{\partial\theta_{k,ij}}pen_2(\boldsymbol{\theta}_{ij})\right| < C_3\sqrt{\log p/n}.$$

*and*

$$\frac{\alpha}{n}\left|\frac{\partial^2}{\partial\theta_{k,ij}\partial\theta_{k',ij}}pen_2(\boldsymbol{\theta}_{ij})\right| < \frac{C_3^2 \log p}{4\alpha p^{\epsilon_0/\alpha}}$$

**Proof**

- For the first derivative, we have

$$\frac{1}{n}\left|\frac{\partial}{\partial\theta_{k,ij}}\text{pen}_2(\boldsymbol{\theta}_{ij})\right| = \frac{1}{n}\frac{\frac{p_1}{v_1^{K+1}}e^{-\|\boldsymbol{\theta}_{ij}\|_1/v_1} + \frac{1-p_1}{v_0^{K+1}}e^{-\|\boldsymbol{\theta}_{ij}\|_1/v_0}}{\frac{p_1}{v_1^K}e^{-\|\boldsymbol{\theta}_{ij}\|_1/v_1} + \frac{1-p_1}{v_0^K}e^{-\|\boldsymbol{\theta}_{ij}\|_1/v_0}}$$

$$= \frac{1}{nv_1} + \frac{\frac{1}{v_0} - \frac{1}{v_1}}{n\left[\frac{p_1 v_0^K}{(1-p_1)v_1^K}e^{\|\boldsymbol{\theta}_{ij}\|_1(1/v_0-1/v_1)} + 1\right]}$$

$$\leqslant \frac{1}{nv_1}\left(1 + \frac{\frac{(1-p_1)v_1^{K+1}}{p_1 v_0^{K+1}}}{e^{\|\boldsymbol{\theta}_{ij}\|_1(1/v_0-1/v_1)}}\right)$$

$$\leqslant \frac{1}{nv_1}(1 + \epsilon_1),$$

where the last line is because

$$\begin{cases} \frac{(1-p_1)v_1^{K+1}}{p_1 v_0^{K+1}} < \epsilon_1 p^{\epsilon_0/\alpha}, \\ e^{\|\boldsymbol{\theta}_{ij}\|_1(1/v_0 - 1/v_1)} > p^{2\epsilon_0/\alpha}. \end{cases}$$

Therefore,

$$\frac{\alpha}{n}\left|\frac{\partial}{\partial\theta_{k,ij}}\mathrm{pen}_2(\boldsymbol{\theta}_{ij})\right| \leqslant \frac{\alpha}{nv_1}\left(1+\epsilon_1\right)$$
$$\leqslant C_3\sqrt{\log p/n}.$$

- For the second derivative, we have

$$\frac{1}{n}\left|\frac{\partial^2}{\partial\theta_{k,ij}\partial\theta_{k',ij}}\mathrm{pen}_2(\boldsymbol{\theta}_{ij})\right| = \frac{\left(\frac{1}{v_0}-\frac{1}{v_1}\right)^2 \frac{p_1 v_0^K}{(1-p_1)v_1^K}e^{-\|\boldsymbol{\theta}_{ij}\|_1(1/v_0-1/v_1)}}{n\left[\frac{p_1 v_0^K}{(1-p_1)v_1^K}e^{-\|\boldsymbol{\theta}_{ij}\|_1(1/v_0-1/v_1)}+1\right]^2}$$

$$\leqslant \frac{\left(\frac{1}{v_0}-\frac{1}{v_1}\right)^2}{n\left[\frac{p_1 v_0^K}{(1-p_1)v_1^K}e^{\|\boldsymbol{\theta}_{ij}\|_1(1/v_0-1/v_1)}+1\right]}$$

$$\leqslant \frac{1}{nv_1^2}\frac{\frac{(1-p_1)v_1^{K+2}}{p_1 v_0^{K+2}}}{e^{\|\boldsymbol{\theta}_{ij}\|_1(1/v_0-1/v_1)}}$$

$$\leqslant \frac{1}{nv_1^2}\frac{\epsilon_1}{p^{\epsilon_0/\alpha}}.$$

Therefore,

$$\frac{\alpha}{n}\left|\frac{\partial^2}{\partial\theta_{k,ij}\partial\theta_{k',ij}}\mathrm{pen}_2(\boldsymbol{\theta}_{ij})\right| \leqslant \frac{\alpha}{nv_1^2}\frac{\epsilon_1}{p^{\epsilon_0/\alpha}}$$

$$\leqslant \frac{\alpha}{n}\frac{n^2}{\alpha^2}\frac{C_3^2}{(1+\epsilon_1)^2}\frac{\log p}{n}\frac{\epsilon_1}{p^{\epsilon_0/\alpha}}$$

$$\leqslant \frac{C_3^2\log p}{4\alpha p^{\epsilon_0/\alpha}}.$$

■

# 3 Computation: an EM algorithm

In this section, we present an EM algorithm for the computation. We treat $\Gamma = (\gamma_{ij})$ as latent variables and estimate $\Theta$ by applying the Expectation step (E-step) and Maximization step (M-step) iteratively.

## 3.1 E-step

In the E-step, we compute the $Q$ function defined as the expectation of the full log likelihood with respect to the joint posterior distribution of the latent variables, $\Gamma$, given $\boldsymbol{Y}$ and $\Theta^{(t)}$, the current estimate of $\Theta$:

$$Q(\Theta \mid \Theta^{(t)}) = \mathbb{E}_{\Gamma\mid\Theta^{(t)},\boldsymbol{Y}} \log\left(p(\boldsymbol{Y}\mid\Theta,\Gamma)^{\frac{1}{\alpha}}p(\Theta,\Gamma)\right)$$

$$= \mathbb{E}_{\Gamma\mid\Theta^{(t)},\boldsymbol{Y}}\left(-\sum_{i<j}\sum_{k=1}^{K}\left[r_{k,ij}\frac{|\theta_{k,ij}|}{v_1}+(1-r_{k,ij})\frac{|\theta_{k,ij}|}{v_0}\right]\right)$$

$$+ \frac{1}{\alpha} \log p(\boldsymbol{Y} \mid \boldsymbol{\Theta}) - \sum_{i=1}^{p} \sum_{k=1}^{K} \tau \theta_{k,ii} + C,$$

where C is some constant with respect to $\boldsymbol{\Theta}$.

For any $i < j$, we have

$$p(\gamma_{ij}, \mid \Theta^{(t)}, Y) \propto \left[ p_1 \prod_{k=1}^{K} \mathrm{LP}(\theta_{k,ij}^{(t)}, v_1) \right]^{\gamma_{ij}} \left[ (1 - p_1) \prod_{k=1}^{K} \mathrm{LP}(\theta_{k,ij}^{(t)}, v_0) \right]^{1-\gamma_{ij}},$$

and therefore,

$$\mathbb{E}_{\Gamma \mid \Theta^{(t)}, Y}(\gamma_{ij}) = \frac{1}{1 + \frac{1-p_1}{p_1} (\frac{v_1}{v_0})^K \exp\left\{ -(\frac{1}{v_0} - \frac{1}{v_1}) \| \boldsymbol{\theta}_{ij}^{(t)} \|_1 \right\}} = w(\boldsymbol{\theta}_{ij}^{(t)}). \tag{3.1}$$

Thus, we have the $Q$ function as

$$Q(\boldsymbol{\Theta} \mid \boldsymbol{\Theta}^{(t)}) = \sum_{k=1}^{K} \left\{ \frac{n_k}{2\alpha} \log \det(\Theta_k) - \frac{n_k}{2\alpha} \mathrm{tr}(S_k \Theta_k) - \sum_{i=1}^{p} \tau \theta_{k,ii} \right.$$
$$\left. - \sum_{i<j} \left[ w(\boldsymbol{\theta}_{ij}^{(t)}) \frac{|\theta_{k,ij}|}{v_1} + \left( 1 - w(\boldsymbol{\theta}_{ij}^{(t)}) \right) \frac{|\theta_{k,ij}|}{v_0} \right] \right\} + C, \tag{3.2}$$

where C is some constant with respect to $\boldsymbol{\Theta}$.

## 3.2 M-step

In the M-step, we update the estimate of $\boldsymbol{\Theta}$ by maximizing the $Q$ function (3.2) under the constraints that $\Theta_k > 0$ and $\|\Theta_k\|_2 \leqslant B$ for $k = 1, \ldots, K$. It is worth noting that $Q(\boldsymbol{\Theta} \mid \boldsymbol{\Theta}^{(t)})$ is a summation of $K$ terms with each to be a function of $\Theta_k$ only. Therefore, our constrained optimization problem can be paralleled by maximizing

$$\frac{n_k}{2\alpha} \log \det(\Theta_k) - \frac{n_k}{2\alpha} \mathrm{tr}(S_k \Theta_k) + \sum_{i=1}^{p} \tau \theta_{k,ii} - \sum_{i<j} \left[ \frac{1}{v_0} + \left( \frac{1}{v_1} - \frac{1}{v_0} \right) w(\boldsymbol{\theta}_{ij}^{(t)}) \right] |\theta_{k,ij}|, \tag{3.3}$$

under the constraints of $\Theta_k > 0$ and $\|\Theta_k\|_2 \leqslant B$ for $k = 1, \ldots, K$. For each sub-problem of maximizing (3.3), we adopt the BAGUS algorithm [2] and update $\Theta_k$ sequentially in the same column by column fashion. The time complexity of our algorithm is thus $O(p^3)$, the same as the state-of-the-art algorithms for graphical Lasso problems. In addition, our final estimates $\hat{\Theta}_k$'s are all symmetric and positive definite as guaranteed in [2].

The maximization of (3.3) can be interpreted as a graphical Lasso problem with adaptive shrinkage where the weight of the penalty is a decreasing function of $\|\boldsymbol{\theta}_{ij}^{(t)}\|_1$. Therefore, for a group with large signals, every entry within the group would not be shrunk heavily. This adaptive shrinkage is advantageous compared to individual estimation methods, because it takes into account the group information to adjust the adaptive weights of penalization for each entry. For example, when small signals are presented in a group with large signals, their estimates will get shrunk less compared to individual estimation methods.

## 3.3 Algorithm

The algorithm is summarized as Algorithm 1. The outputs from the algorithm are the posterior inclusion probability matrix of common structures $P$ and the estimates of the $K$ precision matrices $\Theta_k$'s. For each entry of $P$, we can threshold a pre-specified level, e.g., $0.5$, to get an estimate of the sparsity structure.

---

**Algorithm 1** Joint estimation of multiple graphical models.

---

**repeat**
    Calculate $P = (w(\boldsymbol{\theta}_{ij}^{(t)}))$ using (3.1)
    **for** $k = 1$ to $K$ **do**
        Update $\Theta_k$ by maximizing (3.3)
    **end for**
**until** Convergence
Return $P$ and $\Theta_k, k = 1, \dots, K$

---