[Reviews · NeurIPS 2019]

Reviewer 1



The approach outlines in the paper is well-grounded on a theoretical basis. However, it would be extremely helpful to the reader to explain some of the conditions that allow for theoretical justification. It is not immediately clear if the conditions can be relaxed any further while retaining the same results. The estimation of sparsity structure is an important problem in graphical models.

Reviewer 2



Update after author response: I thank the authors for their very clear response. I think their response very well addresses many of the criticisms that were brought up in the reviews. I especially think that the addition of runtimes will be beneficial to the paper, as this can be a big issue in graphical model learning. On the various technical conditions: I think that providing more intuition for the technical conditions will be great (which the authors address in their response). I think the paper could be made even stronger by adding a discussion of how these conditions compare to those in the literature. ----------------------------------------------------------------------------------------- This paper is focused on estimating multiple Gaussian graphical models. The specific focus is on the setting when we have observations from multiple graphical models with the same underlying sparsity pattern in their precision matrices. The authors propose a point estimator to recover both the individual precision matrices and the underlying sparsity pattern. The authors prove a number of theoretical properties about their method and show that it works well empirically, even when their theoretical assumptions are violated. Clarity: the paper is well written and easy to understand. Originality: The ideas of grouped graphical model estimation and using Bayesian methods in this area are not new. The specific estimator the authors propose and their analysis of it seem to be new. Quality: The theoretical analysis of their method carefully states their assumptions and conclusions. In their empirical work, I appreciated that the authors empirically investigate what happens when their assumption of group structure is violated. I do think that the experiments could be made more convincing, though. First, the authors do not include runtimes for their method, so it is hard to say if the non-huge gains in accuracy are worth the computation cost. Second, it would be good to know how the hyperparameters of the competing methods were selected. Significance: I am not particularly familiar with the graphical model estimation literature, but the authors seem to have provided a good theoretical contribution over other work in the area. My main complaint in this area is that I do not think the characterization of their estimator as "Bayesian" is particularly significant. What is ultimately reported is a constrained MAP estimate that is then thresholded to obtain a sparsity pattern. In contrast, the two Bayesian works that the authors cite in this space ([18] and [22]) actually propose samplers to recover the full posterior. Additionally, it is hard to say that the author's prior has much meaning, given that it places zero probability mass on the sparse solutions of interest. I think this is sort of equivalent to saying the Lasso is a "Bayesian" method. Small things: - In condition A1, should clarify with parentheses whether log p / n < eta means log(p) / n < eta or log(p/n) < eta.

Reviewer 3



This paper proposes a Bayesian group regularization method to jointly estimate the multiple graphical models. The method is based on the spike-and-slab lasso prior. The convergence rate and structure recovery guarantee are provided. Experimental results show the advantage of the proposed method. Overall, the work is solid and interesting. However, I have a few concerns and questions. (1) Why does the training of the model use an EM procedure? Why not optimize the objective at (2.4)? (2) The paper claims the proposed approach is Bayesian. Then why does the paper not compare with other Bayesian approaches to learn multiple CGMs, such as the work [8] and [12] cited in the paper? If there are no special reasons, I strongly recommend the authors to supplement the experiments.

Reviewer 4



I updated my score to 6 since the author's provided some nice explanations in their author response including timing estimates and Bayesian characterization. ----- Original review ----- Originality: Overall the paper seems weak in originality given the amount of previous work on similar problems. The Bayesian viewpoint is somewhat interesting but the paper suggests using a MAP estimate anyways, which is equivalent to a penalized likelihood optimization. Quality: The theoretical results seem reasonable but not particularly surprising given the amount of work on sparse estimation including multiple precision matrices. The empirical results are slightly more promising but given that only one real dataset is used, I'm concerned that this won't generalize to other datasets. Maybe other datasets from [5, 11, 12, 15] could be used? Clarity: Overall, it would be beneficial if the paper included more insights and intuitions about the theorems. Why are the theoretical results particularly surprising or interesting? Additionally, the theorems seem to require many definitions and conditions, which makes it difficult to separate out what's new. Could these be simplified or reorganized for better clarity? Significance: Overall, the paper seems relatively incremental since many others have explored the general problem of learning multiple graphical models together including some theoretical results. The theoretical results are slightly stronger than previous papers but I'm not sure this work would impact ML practitioners or inspire new theoretical work. Other note: What is the computational complexity of this method compared to others? This is always an important thing to know in these situations.

[Author Response · NeurIPS 2019]

We thank the reviewers for the insightful comments. Below are our responses to each reviewer's comments.

**Reviewer #1:**

*Explanation on the Technical Conditions.* We already provided some explanations and intuitions on the technical
conditions but these explanations were buried in the text without concrete references to the corresponding conditions:
Line 117-121 for condition (A.1) and (A.2); Line 126 for condition (A.3); Line 147-156 for conditions (i) - (iv) on the
hyperparameters and sample size; Line 171-174 on the minimal signal condition (v). In the revision, we will re-organize
them and make concrete references to make the discussion on the conditions clearer. We will also provide more detailed
explanations of the technical conditions (A.1) - (A.3). While we think it is possible to relax some of our conditions, it
would require some new technical machinery which is beyond the scope of the current article.

*Choice of Classes and Its Influence on Estimation.* In our setting, the classes are pre-specified by side information or
domain knowledge; the same assumption is held in existing work on multiple graphical models. For estimation accuracy,
the rate of convergence for estimating $\Theta_k$ (the precision matrix for the $k$-th class) is $O_p(\sqrt{(\log p)/n})$ regardless of the
class size $K$, except that to achieve this rate, the sample size $n$ has a lower bound $M_0^2 \max(d^2, K) \log p$ that depends
on $K$ (condition (ii) in Theorem 2). For our bike-sharing data, it is natural to expect the precision matrix changes over
year due to annual policy decisions, economic conditions, and other aspects of the business. So classes are chosen to be
different years.

**Reviewer #2:**

*Computational Times.* In the revision, we shall add the following table on average computational times of all the
methods based on 10 replications. The computational time of our method is comparable to the competitors except the
Pooled method, which restrictively assumes the same precision matrix for all classes and has much worse performance
compared to our method. Therefore, our method is competitive even after considering the runtimes.

| | Nearest-neighbor Network | | | Scale-free Network | | |
|---|---|---|---|---|---|---|
| | $\rho = 0$ | $\rho = 0.25$ | $\rho = 0.5$ | $\rho = 0$ | $\rho = 0.25$ | $\rho = 0.5$ |
| Our method ($\alpha = 1$) | 3.667(0.040) | 3.645(0.087) | 3.552(0.026) | 3.556(0.037) | 3.545(0.030) | 3.537(0.033) |
| Our method ($\alpha = n$) | 7.792(0.456) | 4.596(0.643) | 3.597(0.049) | 5.285(2.623) | 3.600(0.025) | 3.578(0.023) |
| BAGUS | 3.635(0.023) | 3.572(0.027) | 3.547(0.021) | 3.553(0.012) | 3.546(0.022) | 3.534(0.018) |
| Pooled | 1.211(0.010) | 1.178(0.013) | 1.169(0.008) | 1.184(0.015) | 1.173(0.008) | 1.168(0.010) |
| GGL | 8.715(0.314) | 8.034(0.689) | 5.482(1.528) | 8.086(0.262) | 6.139(0.678) | 3.074(0.270) |

*Selection of Hyperparameters.* For all methods, we use a grid search to select the set of hyperparamters that minimizes
BIC. For BAGUS and Pooled methods, we follow the same tuning procedure in [10] and tune the spike and slab
variances $(v_0, v_1)$ with $v_0 = (0.25, 0.5, 0.75, 1) \times \sqrt{1/(n \log p)}$ and $v_1 = (2.5, 5, 7.5, 10) \times \sqrt{1/(n \log p)}$. For GGL,
we tune the two penalty parameters $(\lambda_1, \lambda_2)$ as in [5] with $\lambda_1 = (0.1, 0.2, \ldots, 1)$ and $\lambda_2 = (0.1, 0.3, 0.5)$. We shall
add these details in the revision.

*Bayesian Characterization and Prior.* Our model is indeed a Bayesian model although we emphasize on the MAP
estimator corresponding to our Bayesian model. i) Our model is formulated from a Bayesian perspective with a
continuous spike and slab prior distribution. Although this prior does not directly place mass on sparse solutions, the
latent binary indicators $\gamma_{ij}$ introduced can distinguish between"signal" and "noise". This is a common technique used
in the Bayesian literature to avoid the computational bottleneck of degenerate priors. ii) Although we use the MAP
estimator that also has a penalization interpretation, it is not the only goal in our inference. Without the Bayesian
machinery in the paper, we cannot extract the posterior inclusion probabilities for structure recovery using (2.5)
and provide consequent strong guarantees for graph selection in Theorem 3. iii) For scalability, we compute the
MAP estimator instead of sampling from the full posterior. Full posterior sampling for high-dimensional GGMs is
computationally expensive, for example, in the two Bayesian papers we cited [18, 22], the dimension $p$ in all empirical
studies is less than 22. Further, although MCMC-based samplers are proposed to recover the full posterior in [18], for
structure recovery, only one model is reported based on the same thresholding procedure as our eq (2.6).

*Notations in (A.1).* The notation is $(\log p)/n$. We shall add the parentheses in the revision.

**Reviewer #3:**

*EM vs Direct Optimization.* We agree that it is possible to directly optimize (2.4) after we show that the restricted
optimization problem is strictly convex in Theorem 1. We use the proposed EM algorithm due to the following two
reasons: 1) the E-step provides estimates of posterior inclusion probabilities (2.5), which will be used for structure
recovery; 2) the computational complexity of our EM algorithm is $O(p^3)$, which is already as efficient as the state-of-
the-art algorithms for Graphical Lasso problems [9, 10].

*Comparison with Other Bayesian Alternatives.* We did not compare with Bayesian approaches [18] and [22] since
their MCMC samplers are not scalable with large $p$. Specifically, the largest $p$ handled in [18] and [22] is only 20
and 22, respectively, and [22] states in their Section 6.6 that their method is not scalable to large $p$ and reports the
average computational time to be $(12.4 \pm 0.5)$ hours for its simulation design with $K = 2$, $n_1 = n_2 = 50$, and $p = 20$.
Therefore, they are not computationally manageable to our simulation designs and real data application with large $p$.

[Meta-Review · NeurIPS 2019]

All the reviewers agree that the paper presents an interesting result and is nicely written. Please incorporate reviewers' feedback. Congratulations on a nice result.